# Neural Path Features and Neural Path Kernel : Understanding the role of gates in deep learning

**Chandrashekar Lakshminarayanan**[*] **and Amit Vikram Singh,**[*]
Indian Institute of Technology Palakkad
chandru@iitpkd.ac.in, amitkvikram@gmail.com

## Abstract

Rectified linear unit (ReLU) activations can also be thought of as *gates*, which, either pass or stop their pre-activation input when they are *on* (when the pre-activation input is positive) or *off* (when the pre-activation input is negative) respectively. A deep neural network (DNN) with ReLU activations has many gates, and the on/off status of each gate changes across input examples as well as network weights. For a given input example, only a subset of gates are *active*, i.e., on, and the sub-network of weights connected to these active gates is responsible for producing the output. At randomised initialisation, the active sub-network corresponding to a given input example is random. During training, as the weights are learnt, the active sub-networks are also learnt, and could hold valuable information.

In this paper, we analytically characterise the role of gates and active sub-networks in deep learning. To this end, we encode the on/off state of the gates for a given input in a novel *neural path feature* (NPF), and the weights of the DNN are encoded in a novel *neural path value* (NPV). Further, we show that the output of network is indeed the inner product of NPF and NPV. The main result of the paper shows that the *neural path kernel* associated with the NPF is a fundamental quantity that characterises the information stored in the gates of a DNN. We show via experiments (on MNIST and CIFAR-10) that in standard DNNs with ReLU activations NPFs are learnt during training and such learning is key for generalisation. Furthermore, NPFs and NPVs can be learnt in two separate networks and such learning also generalises well in experiments. In our experiments, we observe that almost all the information learnt by a DNN with ReLU activations is stored in the gates - a novel observation that underscores the need to investigate the role of the gates in DNNs.

## 1   Introduction

We consider deep neural networks (DNNs) with rectified linear unit (ReLU) activations. A special property of the ReLU activation (denoted by $\chi$) is that its output can be written as a product of its pre-activation input, say $q \in \mathbb{R}$ and a gating signal, $G(q) = \mathbb{1}_{\{q>0\}}$, i.e., $\chi(q) = q \cdot G(q)$. While the weights of a DNN remain the same across input examples, the $1/0$ state of the gates (or simply gates) change across input examples. For each input example, there is a corresponding *active sub-network* consisting of those gates which are $1$, and the weights which pass through such gates. This active sub-network can be said to hold the memory for a given input, i.e., only those weights that pass through such active gates contribute to the output. In this viewpoint, at random initialisation of the weights, for a given input example, a random sub-network is active and produces a random output. However, as the weights change during training (say using gradient descent), the gates change, and hence the active sub-networks corresponding to the various input examples also change. At the end

---

[*]Equal Contribution

of training, for each input example, there is a learnt active sub-network, and produces a learnt output. Thus, the gates of a trained DNN could potentially contain valuable information.

We focus on DNNs with ReLU activations. The goal and claims in this paper are stated below.

Goal : *To study the role of the gates in DNNs trained with gradient descent (GD).*
Claim I (Section 5) : *Active sub-networks are fundamental entities.*
Claim II (Section 6) : *Learning of the active sub-networks improves generalisation.*

Before we discuss the claims in terms of our novel technical contributions in Section 1.2, we present the background of *neural tangent* framework in Section 1.1.

**Notation:** We denote the set $\{1, \ldots, n\}$ by $[n]$. For $x, y \in \mathbb{R}^m$, $\langle x, y \rangle = x^\top y$. The dataset is denoted by $(x_s, y_s)_{s=1}^n \in \mathbb{R}^{d_{\text{in}}} \times \mathbb{R}$. For an input $x \in \mathbb{R}^{d_{\text{in}}}$, the output of the DNN is denoted by $\hat{y}_\Theta(x) \in \mathbb{R}$, where $\Theta \in \mathbb{R}^{d_{\text{net}}}$ are the weights. We use $\theta \in \Theta$ to denote a single arbitrary weight, and $\partial_\theta(\cdot)$ to denote $\frac{\partial(\cdot)}{\partial \theta}$. We use $\nabla_\Theta(\cdot)$ to denote the gradient of $(\cdot)$ with respect to the network weights. We use vectorised notations $y = (y_s, s \in [n]), \hat{y}_\Theta = (\hat{y}_\Theta(x_s), s \in [n]) \in \mathbb{R}^n$ for the true and predicted outputs and $e_t = (\hat{y}_{\Theta_t} - y) \in \mathbb{R}^n$ for the error in the prediction.

## 1.1 Background: Neural Tangent Feature and Neural Tangent Kernel

The neural tangent machinery was developed in some of the recent works [8, 1, 3, 5] to understand optimisation and generalisation in DNNs trained using GD. For an input $x \in \mathbb{R}^{d_{\text{in}}}$, the *neural tangent feature* (NTF) is given by $\psi_{x,\Theta} = \nabla_\Theta \hat{y}_\Theta(x) \in \mathbb{R}^{d_{\text{net}}}$, i.e., the gradient of the network output with respect to its weights. The *neural tangent kernel* (NTK) matrix $K_\Theta$ on the dataset is the $n \times n$ Gram matrix of the NTFs of the input examples, and is given by $K_\Theta(s, s') = \langle \psi_{x_s, \Theta}, \psi_{x_{s'}, \Theta} \rangle, s, s' \in [n]$.

**Proposition 1.1** (**Lemma 3.1** Arora et al. [2019a]). *Consider the GD procedure to minimise the squared loss $L(\Theta) = \frac{1}{2} \sum_{s=1}^n (\hat{y}_\Theta(x_s) - y_s)^2$ with infinitesimally small step-size given by $\dot{\Theta}_t = -\nabla_\Theta L_{\Theta_t}$. It follows that the dynamics of the error term can be written as $\dot{e}_t = -K_{\Theta_t} e_t$.*

**Prior works** [8, 5, 1, 3] have studied DNNs trained using GD in the so called 'NTK regime', which occurs (under appropriate randomised initialisation) when the width of the DNN approaches infinity. The characterising property of the NTK regime is that as $w \to \infty$, $K_{\Theta_0} \to K^{(d)}$, and $K_{\Theta_t} \approx K_{\Theta_0}$, where $K^{(d)}$ (see (2) in Appendix A) is a deterministic matrix whose superscript $(d)$ denotes the depth of the DNN. Arora et al. [2019a] showed that an infinite width DNN trained using GD is equivalent to a kernel method with the limiting NTK matrix $K^{(d)}$ (and hence enjoys the generalisation ability of the limiting NTK matrix $K^{(d)}$). Further, Arora et al. [2019a] proposed a pure kernel method based on what they call the CNTK, which is the limiting NTK matrix $K^{(d)}$ of an infinite width convolutional neural network (CNN). Cao and Gu [2019] showed a generalisation bound of the form $\tilde{\mathcal{O}}\left(d \cdot \sqrt{y^\top (K^{(d)})^{-1} y / n}\right)^2$ in the NTK regime.

**Open Question:** Arora et al. [2019a] reported a $5\% - 6\%$ performance gain of finite width CNN (not operating in the NTK regime) over the exact CNTK corresponding to infinite width CNN, and inferred that the study of DNNs in the NTK regime cannot fully explain the success of practical neural networks yet. Can we explain the reason for the performance gain of CNNs over CNTK?

## 1.2 Our Contributions

To the best of our knowledge, we are the first to analytically characterise the role played by the gates and the active sub-networks in deep learning as presented in the 'Claims I and II'. The key contributions can be arranged into three landmarks as described below.

• The first step involves breaking a DNN into individual paths, and each path again into gates and weights. To this end, we encode the states of the gates in a novel *neural path feature* (NPF) and the weights in a novel *neural path value* (NPV) and express the output of the DNN as an inner product of NPF and NPV (see Section 2). In contrast to NTF/NTK which are *first-order* quantities (based on derivatives with respect to the weights), NPF and NPV are *zeroth-order* quantities. The kernel matrix associated to the NPFs namely the *neural path kernel* (NPK) matrix $H_\Theta \in \mathbb{R}^{n \times n}$ has a special structure, i.e., it can be written as a *Hadamard* product of the input Gram matrix, and a correlation

matrix $\Lambda_\Theta \in \mathbb{R}^{n \times n}$, where $\Lambda_\Theta(s, s')$ is proportional to the total number of paths in the sub-network that is active for both input examples $s, s' \in [n]$. With the $\Lambda_\Theta$ matrix we reach our first landmark.

• Second step is to characterise performance of the gates and the active sub-networks in a 'stand alone' manner. To this end, we consider a new idealised setting namely the fixed NPF (FNPF) setting, wherein, the NPFs are fixed (i.e., held constant) and only the NPV is learnt via gradient descent. In this setting, we show that (see Theorem 5.1), in the limit of infinite width and under randomised initialisation the NTK converges to a matrix $K_{\mathrm{FNPF}}^{(d)} = \text{constant} \times H_{\mathrm{FNPF}}$, where $H_{\mathrm{FNPF}} \in \mathbb{R}^{n \times n}$ is the NPK matrix corresponding to the fixed NPFs. $K^{(d)}$ matrix of Jacot et al. [2018], Arora et al. [2019a], Cao and Gu [2019] becomes the $K_{\mathrm{FNPF}}^{(d)}$ matrix in the FNPF setting, wherein, we initialise the NPV statistically independent of the fixed NPFs (see Theorem 5.1). With Theorem 5.1, we reach our second landmark, i.e. we justify "Claim I", that active sub-networks are fundamental entities, which follows from the fact that $H_{\mathrm{FNPF}} = \Sigma \odot \Lambda_{\mathrm{FNPF}}$, where $\Lambda_{\mathrm{FNFP}}$ corresponds to the fixed NPFs.

• Third step is to show experimentally that sub-network learning happens in practice (see Section 6). We show that in finite width DNNs with ReLU activations, NPFs are learnt continuously during training, and such learning improves generalisation. We observe that fixed NPFs obtained from the initial stages of training generalise poorly than CNTK (of Arora et al. [2019a]), whereas, fixed NPFs obtained from later stages of training generalise better than CNTK and generalise as well as standard DNNs with ReLU. This throws light on the open question in Section 1.1, i.e., the difference between the NTK regime and the finite width DNNs is perhaps due to NPF learning. In finite width DNNs, NPFs are learnt during training and in the NTK regime no such feature learning happens during training. Since the NPFs completely encode the information pertaining to the active sub-networks, we complete our final landmark namely justification of "Claim II".

## 2 Neural Path Feature and Kernel: Encoding Gating Information

First step in understanding the role of the gates is to explicitly *encode* the $1/0$ states of the gates. The gating property of the ReLU activation allows us to express the output of the DNN as a summation of the contribution of the individual paths, and paves a natural way to encode the $1/0$ states of the gates *without loss of information*. The contribution of a path is the product of the signal at its input node, the weights in the path and the gates in the path. For an input $x \in \mathbb{R}^{d_{\mathrm{in}}}$, and weights $\Theta \in \mathbb{R}^{d_{\mathrm{net}}}$, we encode the gating information in a novel *neural path feature* (NPF), $\phi_{x,\Theta} \in \mathbb{R}^P$ and encode the weights in a novel *neural path value* (NPV) $v_\Theta \in \mathbb{R}^P$, where $P$ is the total number of paths. The NPF co-ordinate of a path is the product of the signal at its input node and the gates in the path. The NPV co-ordinate of a path is the product of the weights in the paths. The output is then given by

$$\hat{y}_\Theta(x) = \langle \phi_{x,\Theta}, v_\Theta \rangle, \tag{1}$$

where $\phi_{x,\Theta}$ can be seen as the *hidden features* which along with $v_\Theta$ are learnt by gradient descent.

### 2.1 Paths, Neural Path Feature, Neural Path Value and Network Output

We consider fully-connected DNNs with '$w$' hidden units per layer and '$d-1$' hidden layers. $\Theta \in \mathbb{R}^{d_{\mathrm{net}}}$ are the network weights, where $d_{\mathrm{net}} = d_{\mathrm{in}}w + (d-2)w^2 + w$. The information flow is shown in Table 1, where $\Theta(i, j, l)$ is the weight connecting the $j^{th}$ hidden unit of layer $l-1$ to the $i^{th}$ hidden unit of layer $l$. Further, $\Theta(\cdot, \cdot, 1) \in \mathbb{R}^{w \times d_{\mathrm{in}}}, \Theta(\cdot, \cdot, l) \in \mathbb{R}^{w \times w}, \forall l \in \{2, \ldots, d-1\}, \Theta(\cdot, \cdot, d) \in \mathbb{R}^{1 \times w}$.

| Input Layer | : | $z_{x,\Theta}(0)$ | = | $x$ |
|---|---|---|---|---|
| Pre-Activation Input | : | $q_{x,\Theta}(i, l)$ | = | $\sum_j \Theta(i, j, l) \cdot z_{x,\Theta}(j, l-1)$ |
| Gating Values | : | $G_{x,\Theta}(i, l)$ | = | $\mathbb{1}_{\{q_{x,\Theta}(i,l)>0\}}$ |
| Hidden Layer Output | : | $z_{x,\Theta}(i, l)$ | = | $q_{x,\Theta}(i, l) \cdot G_{x,\Theta}(i, l)$ |
| Final Output | : | $\hat{y}_\Theta(x)$ | = | $\sum_{j \in [w]} \Theta(1, j, d-1) \cdot z_{x,\Theta}(j, d-1)$ |

Table 1: Here, $l \in [d-1], i \in [w], j \in [d_{\mathrm{in}}]$ for $l=1$ and $j \in [w]$ for $l=2, \ldots, d-1$.

**Paths:** A path starts from an input node, passes through exactly one weight and one hidden node in each layer and ends at the output node. We have a total of $P = d_{\mathrm{in}}w^{(d-1)}$ paths. We assume that there is a natural enumeration of the paths, and denote the set of all paths by $[P]$. Let $\mathcal{I}_l : [P] \to$

$[w], l = 1, \ldots, d-1$ provide the index of the hidden unit through which a path $p$ passes in layer $l$, and $\mathcal{I}_0 \colon [P] \to [d_{\text{in}}]$ provides the input node, and $\mathcal{I}_d(p) = 1, \forall p \in [P]$.

**Definition 2.1.** *Let $x \in \mathbb{R}^{d_{in}}$ be the input to the DNN. For this input,*

*(i) The activity of a path $p$ is given by :* $A_\Theta(x, p) \overset{def}{=} \Pi_{l=1}^{d-1} G_{x,\Theta}(\mathcal{I}_l(p), l)$.

*(ii) The neural path feature (NPF) is given by :* $\phi_{x,\Theta} \overset{def}{=} (x(\mathcal{I}_0(p)) A_\Theta(x, p), p \in [P]) \in \mathbb{R}^P$.

*(iii) The neural path value (NPV) is given by :* $v_\Theta \overset{def}{=} \left( \Pi_{l=1}^d \Theta(\mathcal{I}_l(p), \mathcal{I}_{l-1}(p), l), p \in [P] \right) \in \mathbb{R}^P$.

**Remark:** A path $p$ is active if all the gates in the paths are on.

**Proposition 2.1.** *The output of the network can be written as an inner product of the NPF and NPV, i.e.,* $\hat{y}_\Theta(x) = \langle \phi_{x,\Theta}, v_\Theta \rangle = \sum_{p \in [P]} x(\mathcal{I}_0(p)) A_\Theta(x, p) v_\Theta(p)$.

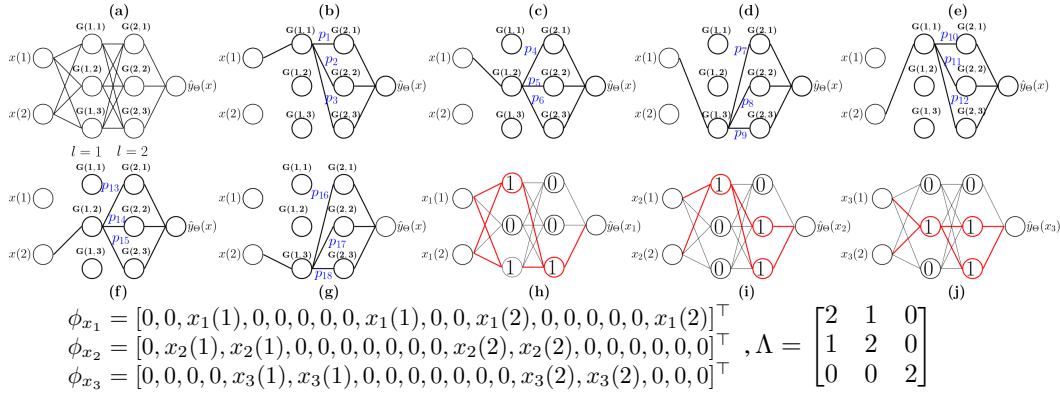

$$\phi_{x_1} = [0, 0, x_1(1), 0, 0, 0, 0, 0, x_1(1), 0, 0, x_1(2), 0, 0, 0, 0, 0, x_1(2)]^\top$$
$$\phi_{x_2} = [0, x_2(1), x_2(1), 0, 0, 0, 0, 0, 0, 0, 0, x_2(2), x_2(2), 0, 0, 0, 0, 0]^\top \quad , \Lambda = \begin{bmatrix} 2 & 1 & 0 \\ 1 & 2 & 0 \\ 0 & 0 & 2 \end{bmatrix}$$
$$\phi_{x_3} = [0, 0, 0, 0, x_3(1), x_3(1), 0, 0, 0, 0, 0, 0, 0, 0, x_3(2), x_3(2), 0, 0, 0]^\top$$

Figure 1: A toy illustration of gates, paths and active sub-networks. The cartoon (**a**) in the top left corner shows a DNN with 2 hidden layers, 6 ReLU gates $G(l, i), l = 1, 2, i = 1, 2, 3$, 2 input nodes $x(1)$ and $x(2)$ and an output node $\hat{y}_\Theta(x)$. Cartoons (**b**) to (**g**) show the enumeration of the paths $p_1, \ldots, p_{18}$. Cartoons (**h**), (**i**) and (**j**) show hypothetical gates for 3 different hypothetical input examples $\{x_s\}_{s=1}^3 \in \mathbb{R}^2$. In each of the cartoons (**h**), (**i**) and (**j**), the $1/0$ inside the circles denotes the on/off state of the gates, and the bold paths/gates shown in red colour constitute the active sub-network for that particular input example. The NPFs are given by $\phi_x = [x(1)A(x, p_1), \ldots, x(1)A(x, p_9), x(2)A(x, p_{10}), \ldots, x(2)A(x, p_{18})]^\top$. Here, $\Lambda(1, 2) = 1$ because paths $p_3$ and $p_{12}$ are both active for input examples $x_1$ and $x_2$ and the input dimension is 2.

## 2.2 Neural Path Kernel : Similarity based on active sub-networks

**Definition 2.2.** *For input examples $s, s' \in [n]$, define $Act_\Theta(s, s') \overset{def}{=} \{p \in [P] \colon A_\Theta(x_s, p) = A_\Theta(x_{s'}, p) = 1\}$ to be the set of 'active' paths for both $s, s'$ and $\Lambda_\Theta(s, s') \overset{def}{=} \frac{|Act_\Theta(s, s')|}{d_{in}}$.*

**Remark:** Owing to the symmetry of a DNN, the same number of active paths start from any fixed input node. In Definition 2.2, $\Lambda_\Theta$ measures the size of the active sub-network as the total number of active paths starting from any fixed input node. For examples $s, s' \in [n], s \neq s'$, $\Lambda_\Theta(s, s)$ is equal to the size of the sub-network active for $s$, and $\Lambda_\Theta(s, s')$ is equal to the size of the sub-network active for both $s$ and $s'$. For an illustration of NPFs and $\Lambda$ please see Figure 1.

**Lemma 2.1.** *Let $H_\Theta \in \mathbb{R}^{n \times n}$ be the NPK matrix, whose entries are given by $H_\Theta(s, s') \overset{def}{=} \langle \phi_{x_s, \Theta}, \phi_{x_{s'}, \Theta} \rangle, s, s' \in [n]$. Let $\Sigma \in \mathbb{R}^{n \times n}$ be the input Gram matrix with entires $\Sigma(s, s') = \langle x_s, x_{s'} \rangle, s, s' \in [n]$. It follows that $H_\Theta = \Sigma \odot \Lambda_\Theta$, where $\odot$ is the Hadamard product.*

# 3 Dynamics of Gradient Descent with NPF and NPV Learning

In Section 2, we mentioned that during gradient descent, the DNN is learning a relation $\hat{y}_\Theta(x) = \langle \phi_{x,\Theta}, v_\Theta \rangle$, i.e., both the NPFs and the NPV are learnt during gradient descent. In this section, we

connect the newly defined quantities, i.e, NPFs and NPV to the NTK matrix $K_\Theta$ (see Proposition 3.1), and re-write the gradient descent dynamics in Proposition 3.2. In what follows, we use $\Phi_\Theta = (\phi_{x_s,\Theta}, s \in [n]) \in \mathbb{R}^{P \times n}$ to denote the NPF matrix.

## 3.1 Dynamics of NPFs and NPV

**Definition 3.1.** *The gradient of the NPV of a path $p$ is defined as $\varphi_{p,\Theta} \overset{def}{=} (\partial_\theta v_\Theta(p), \theta \in \Theta) \in \mathbb{R}^{d_{net}}$.*

**Remark:** The change of the NPV is given by $\dot{v}_{\Theta_t}(p) = \langle \varphi_{p,\Theta_t}, \dot{\Theta}_t \rangle$, where $\dot{\Theta}_t$ is the change of the weights. We now collect the gradients $\varphi_{p,\Theta}$ of all the paths to define a *value tangent kernel* (VTK).

**Definition 3.2.** *Let $\nabla_\Theta v_\Theta$ be a $d_{net} \times P$ matrix of NPV derivatives given by $\nabla_\Theta v_\Theta = (\varphi_{p,\Theta}, p \in [P])$. Define the VTK to be the $P \times P$ matrix given by $\mathcal{V}_\Theta = (\nabla_\Theta v_\Theta)^\top (\nabla_\Theta v_\Theta)$.*

**Remark:** An important point to note here is that the VTK is a quantity that is dependent only on the weights. To appreciate the same, consider a deep linear network (DLN) [13, 4] which has identity activations, i.e., all the gates are 1 for all inputs, and weights. For a DLN and DNN with identical network architecture (i.e., $w$ and $d$), and identical weights, $\mathcal{V}_\Theta$ is also identical. Thus, $\mathcal{V}_\Theta$ is the gradient based information that excludes the gating information.

The NPFs changes at those time instants when any one of the gates switches from 1 to 0 or from 0 to 1. In the time between two such switching instances, NPFs of all the input examples in the dataset remain the same, and between successive switching instances, the NPF of at least one of the input example in the dataset changes. In what follows, in Proposition 3.2 we re-write Proposition 1.1 taking into account the switching instances which we define in Definition 3.3.

**Definition 3.3.** *Define a sequence of monotonically increasing time instants $\{T_i\}_{i=0}^\infty$ (with $T_0 = 0$) to be 'switching' instants if $\phi_{x_s,\Theta_t} = \phi_{x_s,\Theta_{T_i}}, \forall s \in [n], \forall t \in [T_i, T_{i+1}), i \geq 0$, and $\forall i \geq 0$, there exists $s(i) \in [n]$ such that $\phi_{x_{s(i)},\Theta_{T_i}} \neq \phi_{x_{s(i)},\Theta_{T_{i+1}}}$.*

## 3.2 Re-writing Gradient Descent Dynamics

**Proposition 3.1.** *The NTK is given by $K_\Theta = \Phi_\Theta^\top \mathcal{V}_\Theta \Phi_\Theta$.*

**Remark:** $K_{\Theta_t}$ changes during training (i) continuously at all $t \geq 0$ due to $\mathcal{V}_{\Theta_t}$, and (ii) at switching instants $T_i, i = 0, \ldots, \infty$ due to the change in $\Phi_{\Theta_{T_i}}$. We now describe the gradient descent dynamics taking into the dynamics of the NPV and the NPFs.

**Proposition 3.2.** *Let $\{T_i\}_{i=0}^\infty$ be as in Definition 3.3. For $t \in [T_i, T_{i+1})$ and small step-size of GD:*

$$
\begin{array}{llll}
\text{Weights Dynamics} & : & \dot{\Theta}_t & = & -\sum_{s=1}^n \psi_{x_s,\Theta_t} e_t(s) \\
\text{NPV Dynamics} & : & \dot{v}_{\Theta_t}(p) & = & \langle \varphi_{p,\Theta_t}, \dot{\Theta}_t \rangle, \forall p \in [P] \\
\text{Error Dynamics} & : & \dot{e}_t & = & -K_{\Theta_t} e_t, \text{ where } K_{\Theta_t} = \Phi_{\Theta_{T_i}}^\top \mathcal{V}_{\Theta_t} \Phi_{\Theta_{T_i}}
\end{array}
$$

**Proposition 3.3.** *Let the maximum and minimum eigenvalues of a real symmetric matrix $A$ be denoted by $\rho_{\max}(A)$ and $\rho_{\min}(A)$. Then, $\rho_{\min}(K_\Theta) \leq \rho_{\min}(H_\Theta)\rho_{\max}(\mathcal{V}_\Theta)$.*

**Remark:** For the NTK to be well conditioned, it is necessary for the NPK to be well conditioned. This is intuitive, in that, the closer two inputs are, the closer are their NPFs, and it is harder to train the network to produce arbitrarily different outputs for such inputs that are very close to one another.

# 4 Deep Gated Networks: Decoupling Neural Path Feature and Value

The next step towards our goal of understanding the role of the gates (and gate dynamics) is the separation of the gates (i.e., the NPFs) from the weights (i.e., the NPV). This is achieved by a deep gated network (DGN) having two networks of identical architecture namely i) a feature network parameterised by $\Theta^f \in \mathbb{R}^{d_{net}^f}$, that holds gating information, and hence the NPFs and ii) a value network that holds the NPVs parameterised by $\Theta^v \in \mathbb{R}^{d_{net}^v}$. As shown in Figure 2, the gates/NPFs are generated in the feature network and are used in the value network. In what follows, we let $\Theta^{DGN} = (\Theta^f, \Theta^v) \in \mathbb{R}^{d_{net}^f + d_{net}^v}$ to denote the combined parameters of a DGN.

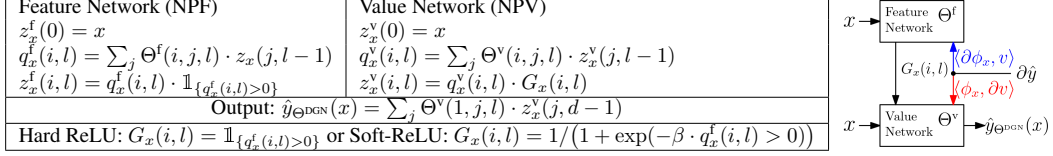

Figure 2: Deep gated network (DGN) setup. The pre-activations $q_x^{\mathrm{f}}(i,l)$ from the feature network are used to derive the gating values $G_x(i,l)$. The range of the indices $i,j,l$ is the same as in Table 1.

**Regimes of a DGN:** We can configure the DGN in four different *regimes* by controlling (i) the trainability of $\Theta^{\mathrm{f}}$, and (ii) the initialisation of $\Theta_0^{\mathrm{f}}$. By setting $\Theta^{\mathrm{f}}$ to be *non-trainable/trainable* we can compare fixed NPFs and NPFs that change during training. By setting $\Theta_0^{\mathrm{f}}$ to be *random/pre-trained* we can compare random NPFs and learnt NPFs. By setting $\Theta_0^{\mathrm{f}} = \Theta_0^{\mathrm{v}}$ we can mimic the initialisation of a standard DNN with ReLU. The four regimes of a DGN are described below (note that in all the regimes $\Theta_0^{\mathrm{v}}$ is randomly initialised, $\Theta^{\mathrm{v}}$ is trainable and $\hat{y}_{\Theta^{\mathrm{DGN}}}$ is the output node).

1. Decoupled Learning of NPF (**DLNPF**): Here, $\Theta^{\mathrm{f}}$ is trainable, and hence the NPFs are learnt in a decoupled manner (as opposed to the standard DNN with ReLU where a single parameter is responsible for learning NPFs and NPV). Here, soft-ReLU gate with $\beta > 0$ is used to ensure gradient flow via feature network. $\Theta_0^{\mathrm{f}}, \Theta_0^{\mathrm{v}}$ are initialised at random and are statistically independent.

2. Fixed Learnt NPF (**FLNPF**): Here $\Theta^{\mathrm{f}}$ is non-trainable, and $\Theta_0^{\mathrm{f}}$ copied from a pre-trained DNN with ReLU (NPFs are learnt). $\Theta_0^{\mathrm{v}}$ are initialised at random and is statistically independent of $\Theta_0^{\mathrm{f}}$.

3. Fixed Random NPF with Independent Initialisation (**FRNPF-II**): Here, $\Theta_0^{\mathrm{f}}, \Theta_0^{\mathrm{v}}$ are initialised at random are statistically independent. Also, $\Theta^{\mathrm{f}}$ is non-trainable, i.e., the NPFs are random and fixed.

4. Fixed Random NPF with Dependent Initialisation (**FRNPF-DI**): Here, the initialisation mimics standard DNN with ReLU, i.e., $\Theta_0^{\mathrm{f}} = \Theta_0^{\mathrm{v}}$ are initialised at random, and $\Theta^{\mathrm{f}}$ is non-trainable.

**Remark:** The DGN and its regimes are idealised models to understand the role of the gates, and not alternate proposals to replace standard DNNs with ReLU activations.

**Proposition 4.1** (Gradient Dynamics in a DGN). *Let* $\psi_{x,\Theta^{\mathrm{DGN}}}^{f} \stackrel{def}{=} \nabla_{\Theta^{\mathrm{f}}}\hat{y}_{\Theta^{\mathrm{DGN}}}(x) \in \mathbb{R}^{d_{net}^{f}}$, $\psi_{x,\Theta^{\mathrm{DGN}}}^{v} \stackrel{def}{=}$ $\nabla_{\Theta^{\mathrm{v}}}\hat{y}_{\Theta^{\mathrm{DGN}}}(x) \in \mathbb{R}^{d_{net}^{v}}$. *Let* $K_{\Theta^{\mathrm{DGN}}}^{v}$ *and* $K_{\Theta^{\mathrm{DGN}}}^{f}$ *be* $n \times n$ *matrices with entries* $K_{\Theta^{\mathrm{DGN}}}^{v}(s,s') = \langle \psi_{x_s,\Theta^{\mathrm{DGN}}}^{v}, \psi_{x_{s'},\Theta^{\mathrm{DGN}}}^{v}\rangle$ *and* $K_{\Theta^{\mathrm{DGN}}}^{f}(s,s') = \langle \psi_{x_s,\Theta^{\mathrm{DGN}}}^{f}, \psi_{x_{s'},\Theta^{\mathrm{DGN}}}^{f}\rangle$. *For infinitesimally small step-size of GD, the error dynamics in a DGN (in the DLNPF and FNPF modes) is given by:*

| Dynamics | Decoupled Learning | | Fixed NPF |
|---|---|---|---|
| Weight | $\dot{\Theta}_t^{\mathrm{v}}$ | $= -\sum_{s=1}^{n}\psi_{x,\Theta_t^{\mathrm{DGN}}}^{v}e_t(s), \dot{\Theta}_t^{\mathrm{f}} = -\sum_{s=1}^{n}\psi_{x,\Theta_t^{\mathrm{f}}}^{f}e_t(s)$ | $\dot{\Theta}_t^{\mathrm{v}}$ same as (DLNFP), $\dot{\Theta}_t^{\mathrm{f}} = 0$ |
| NPF | $\dot{\phi}_{x_s,\Theta_t^{\mathrm{f}}}(p)$ | $= x(\mathcal{I}_0(p))\sum_{\Theta^{\mathrm{f}}\in\Theta^{\mathrm{f}}}\partial_{\Theta^{\mathrm{f}}}A_{\Theta_t^{\mathrm{f}}}(x_s,p)\dot{\theta}_t^{\mathrm{f}}, \forall p \in [P], s \in [n]$ | $\dot{\phi}_{x_s,\Theta_t^{\mathrm{f}}}(p) = 0$ |
| NPV | $\dot{v}_{\Theta_t^{\mathrm{v}}}(p)$ | $= \sum_{\theta^{\mathrm{v}}\in\Theta^{\mathrm{v}}}\partial_{\theta^{\mathrm{v}}}v_{\Theta_t^{\mathrm{v}}}(p)\dot{\theta}_t^{\mathrm{v}}, \forall p \in [P]$ | $\dot{v}_{\Theta_t^{\mathrm{v}}}(p)$ same as DLNPF |
| Error | $\dot{e}_t$ | $= -\left(K_{\Theta^{\mathrm{DGN}}}^{v} + K_{\Theta^{\mathrm{DGN}}}^{f}\right)e_t$ | $\dot{e}_t = -\left(K_{\Theta^{\mathrm{DGN}}}^{v}\right)e_t$ |

**Remark:** The gradient dynamics in a DGN specified in Proposition 4.1 is similar to the gradient dynamics in a DNN specified in Proposition 3.2. Important difference is that (in a DGN) the NTF $\psi_{x,\Theta} = (\psi_{x,\Theta}^{\mathrm{f}}, \psi_{x,\Theta}^{\mathrm{v}}) \in \mathbb{R}^{d_{net}^{f}+d_{net}^{v}}$, wherein, $\psi_{x,\Theta^{\mathrm{DGN}}}^{v} \in \mathbb{R}^{d_{net}^{v}}$ and $\psi_{x,\Theta^{\mathrm{DGN}}}^{f} \in \mathbb{R}^{d_{net}^{f}}$ flow through the value and feature networks respectively. Here, NPF learning is captured explicitly by $\psi^{\mathrm{f}}$ and $K_{\Theta}^{\mathrm{f}}$.

## 5  Learning with Fixed NPFs: Role Of Active Sub-Networks

We now show that in the fixed NPF regime, at randomised initialisation, NTK = const × NPK. Due to the *Hadamard* structure of the NPK (Lemma 2.1) it follows that the active sub-networks are fundamental entities in DNNs (theoretically justifying "Claim I").

**Theorem 5.1.** *Let $H_{FNFP}$ refer to $H_{\Theta_0^f}$. Let (i) $\Theta_0^v \in \mathbb{R}^{d_{net}^v}$ be statistically independent of $\Theta_0^f \in \mathbb{R}^{d_{net}^f}$, and (ii) $\Theta_0^v$ be sampled i.i.d from symmetric Bernoulli over $\{-\sigma, +\sigma\}$. For $\sigma = \frac{\sigma'}{\sqrt{w}}$, as $w \to \infty$,*

$$K_{\Theta_0^{DGN}}^v \to K_{FNPF}^{(d)} = d \cdot \sigma^{2(d-1)} \cdot H_{FNPF}$$

• **Statistical independence** of $\Theta_0^f$ and $\Theta_0^v$ assumed in Theorem 5.1 holds only for the three regimes namely DLNPF, FLNPF, FRNPF-II. In DNN with ReLU (and in FRNPF-DI) $\Theta_0^f = \Theta_0^v$, and hence the assumption in Theorem 5.1 does not capture the conditions at initialisation in a DNN with ReLU. However, it is important to note that the current state-of-the-art analysis for DNNs is in the infinite width regime [8, 1, 3], wherein, the activations undergo only an order of $\frac{1}{\sqrt{w}}$ change during training. Since as $w \to \infty$, $\frac{1}{\sqrt{w}} \to 0$, assuming the NPFs (i.e., gates) to be fixed during training is not a strong one. With fixed NPFs, statistical independence of $\Theta_0^v$ is a natural choice. Also, we do not observe significant empirical difference between the FRNPF-DI and FRNPF-II regimes (see Section 6).

• **Role of active sub-networks:** Due to the statistical independence of $\Theta_0^f$ and $\Theta_0^v$, $K^{(d)}$ in prior works [8, 1, 3] essentially becomes $K_{FNPF}^{(d)}$ in Theorem 5.1. From previous results [1, 3], it follows that as $w \to \infty$, the optimisation and generalisation properties of the fixed NPF learner can be tied down to the infinite width NTK of the FNPF learner $K_{FNPF}^{(d)}$ and hence to $H_{FNPF}$ (treating $d\sigma^{2(d-1)}$ as a scaling factor). We can further breakdown $H_{FNPF} = \Sigma \odot \Lambda_{FNPF}$, where $\Lambda_{FNPF} = \Lambda_{\Theta_0^f}$. This justifies "Claim I", because $\Lambda_{FNPF}$ is the correlation matrix of the active sub-network overlaps. We also observe in our experiments that with the learnt NPFs (i.e., in the FLNPF regime), we can train the NPV without significant loss of performance, a fact that underscores the importance of the NPFs/gates.

• **Scaling factor** '$d$' is due to the '$d$' weights in a path and at $t = 0$ the derivative of the value of a given path with respect any of its weights is $\sigma^{(d-1)}$. In the case of random NPFs obtained by initialising $\Theta_0^f$ at random (by sampling from a symmetric distribution), we expect $\frac{w}{2}$ gates to be 'on' every layer, so $\sigma = \sqrt{\frac{2}{w}}$ is a normalising choice, in that, the diagonal entries of $\sigma^{2(d-1)}\Lambda_{FNPF}(s, s) \approx 1$ in this case.

• Theorem 5.1 can also be applied when the fixed NPFs are obtained from a finite width feature network by a 'repetition trick' (see Appendix C). We also apply Theorem 5.1 to understand the role of width and depth on a pure memorisation task (see Appendix D).

## 6 Experiments: Fixed NPFs, NPF Learning and Verification of Claim II

In this section, we experimentally investigate the role of gates and empirically justify "Claim II", i.e., learning of the active sub-networks improves generalisation. In our framework, since the gates are encoded in the NPFs, we justify "Claim II" by showing that NPF learning improves generalisation. For this purpose, we make use of the DGN setup and its four different regimes. We show that NPF learning also explains the performance gain of finite width CNN over the pure kernel method based on the exact infinite width CNTK. We now describe the setup and then discuss the results.

### 6.1 Setup

**Datasets:** We used standard datasets namely MNIST and CIFAR-10, with categorical cross entropy loss. We also used a 'Binary'-MNIST dataset (with squared loss), which is MNIST with only the two classes corresponding to digits $4$ and $7$, with label $-1$ for digit $4$ and $+1$ for digit $7$.

**Optimisers:** We used stochastic gradient descent (SGD) and *Adam* [10] as optimisers. In the case of SGD, we tried constant step-sizes in the set $\{0.1, 0.01, 0.001\}$ and chose the best. In the case of Adam the we used a constant step size of $3e^{-4}$. In both cases, we used batch size to be $32$.

**Architectures:** For MNIST we used fully connected (FC) architectures with $(w = 128, d = 5)$. For CIFAR-10, we used a *'vanilla' convolutional* architecture namely VCONV and a *convolutional* architecture with *global-average-pooling* (GAP) namely GCONV. GCONV had no pooling, residual connections, dropout or batch-normalisation, and is given as follows: input layer is $(32, 32, 3)$, followed by 4 convolution layers, each with a stride of $(3, 3)$ and channels $64, 64, 128, 128$ respectively. The convolutional layers are followed by GAP layer, and a FC layer with $256$ units, and a soft-max layer to produce the final predictions. VCONV is same as GCONV without the GAP layer.

| Arch | Optimiser | Dataset | FRNPF (II) | FRNPF (DI) | DLNPF ($\beta = 4$) | FLNPF | ReLU |
|------|-----------|---------|------------|------------|---------------------|-------|------|
| FC | SGD | MNIST | $95.85 \pm 0.10$ | $95.85 \pm 0.17$ | $97.86 \pm 0.11$ | $97.10 \pm 0.09$ | $97.85 \pm 0.09$ |
| FC | Adam | MNIST | $96.02 \pm 0.13$ | $96.09 \pm 0.12$ | $\mathbf{98.22 \pm 0.05}$ | $\mathbf{97.82 \pm 0.02}$ | $\mathbf{98.14 \pm 0.07}$ |
| VCONV | SGD | CIFAR-10 | $58.92 \pm 0.62$ | $58.83 \pm 0.27$ | $63.21 \pm 0.07$ | $63.06 \pm 0.73$ | $67.02 \pm 0.43$ |
| VCONV | Adam | CIFAR-10 | $64.86 \pm 1.18$ | $64.68 \pm 0.84$ | $\mathbf{69.45 \pm 0.76}$ | $\mathbf{71.40 \pm 0.47}$ | $\mathbf{72.43 \pm 0.54}$ |
| GCONV | SGD | CIFAR-10 | $67.36 \pm 0.56$ | $66.86 \pm 0.44$ | $\mathbf{74.57 \pm 0.43}$ | $\mathbf{78.52 \pm 0.39}$ | $\mathbf{78.90 \pm 0.37}$ |
| GCONV | Adam | CIFAR-10 | $67.09 \pm 0.58$ | $67.08 \pm 0.27$ | $\mathbf{77.12 \pm 0.19}$ | $\mathbf{79.68 \pm 0.32}$ | $\mathbf{80.32 \pm 0.35}$ |

Table 2: Shows the test accuracy of different NPFs learning settings. Each model is trained close to $100\%$. In each run, the best test accuracy is taken and the table presents values averaged over 5 runs.

## 6.2 Result Discussion

The results are tabulated in Table 2. In what follows, we discuss the key observations.

1. **Decoupling:** There is no significant performance difference between FRNPF-II and FRNPF-DI regimes, i.e., the statistical independence of $\Theta_0^v$ and $\Theta_0^f$ did not affect performance. Also, DLNPF performed better than FRNPF, which shows that the NPFs can also be learnt in a decoupled manner.

2. **Performance gain of CNN over CNTK** can be explained by NPF learning. For this purpose, we look at the performance of GCONV models trained with *Adam* on CIFAR-10 (last row of Table 2). Consider models grouped as $S_1 = \{$FRNPF-DI,FRNPF-II$\}$, $S_2 = \{$CNTK$\}$ that have no NPF learning versus models grouped as $S_3 = \{$ FLNPF, ReLU$\}$ that have either NPF learning during training or a fixed learnt NPF. The group $S_3$ performs better than $S_1$ and $S_2$. Note that, both $S_1$ and $S_3$ are finite width networks, yet, performance of $S_1$ is worse than CNTK, and the performance of $S_3$ is better than CNTK. Thus finite width alone does not explain the performance gain of CNNs over CNTK. Further, all models in group $S_3$ are finite width and also have NPF learning. Thus, finite width together with NPF learning explains the performance gain of CNN over CNTK.

3. **Standard features vs NPFs:** The standard view is that the outputs of the intermediate/hidden layers learn lower to higher level features (as depth proceeds) and the final layer learns a linear model using the hidden features given by the penultimate layer outputs. This view of feature learning holds true for all the models in $S_1$ and $S_3$. However, only NPF learning clearly *discriminates* between the different regimes $S_1, S_2$ and $S_3$. Thus, NPF learning is indeed a unique aspect in deep learning. Further, in the FLNPF regime, using the learnt NPFs and training the NPV from scratch, we can recover the test accuracy. Thus almost all useful information is stored in the gates, a novel observation which underscores the need to further investigate the role of the gates.

4. **Continuous NPF Learning:** The performance gap between FRNPF and ReLU is continuous. We trained a standard ReLU-CNN with GCONV architecture (parameterised by $\bar{\Theta}$) for 60 epochs. We sampled $\bar{\Theta}_t$ at various *stages* of training, where stage $i$ corresponds to $\bar{\Theta}_{10 \times i}, i = 1, \ldots, 6$. For these 6 stages, we setup 6 different FLNPFs, i.e., FLNPF-1 to FLNPF-6. We observe that the performance of FLNPF-1 to FLNPF-6 increases monotonically, i.e., FLNPF-1 performs ($72\%$) better than FRNPF ($67.09\%$), and FLNPF-6 performs as well as ReLU (see left most plot in Figure 3). The performance of CNTK of Arora et al. [2019a] is $77.43\%$. Thus, through its various stages, FLNPF starts from below $77.43\%$ and surpasses to reach $79.43\%$, which implies performance gain of CNN is due to learning of NPFs.

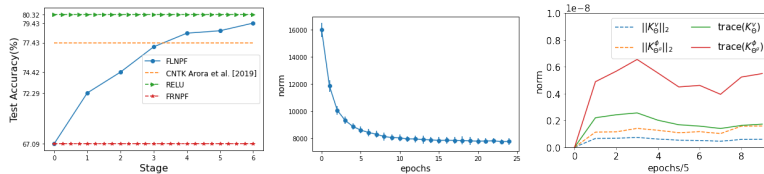

Figure 3: Dynamics of NPF Learning.

5. **Dynamics of active sub-networks during training:** We trained a FC network ($w = 100, d = 5$) on the "Binary"-MNIST dataset. Let $\widehat{H}_{\Theta_t} = \frac{1}{trace(H_{\Theta_t})} H_{\Theta_t}$ be the normalised NPK matrix. For a subset size, $n' = 200$ (100 examples per class) we plot $\nu_t = y^\top (\widehat{H}_{\Theta_t})^{-1} y$, (where $y \in \{-1, 1\}^{200}$ is the labelling function), and observe that $\nu_t$ reduces as training proceeds (see middle plot in Figure 3). Note that, $\nu_t = \sum_{i=1}^{n'} (u_{i,t}^\top y)^2 (\hat{\rho}_{i,t})^{-1}$, where $u_{i,t} \in \mathbb{R}^{n'}$ are the orthonormal eigenvectors of $\widehat{H}_{\Theta_t}$

and $\hat{\rho}_{i,t}, i \in [n']$ are the corresponding eigenvalues. Since $H_{\Theta_t} = \Sigma \odot \Lambda_{\Theta_t}$, we can infer that $\Lambda_{\Theta_t}$ is learnt during training.

6. **How are NPFs learnt?** In order to understand this, in the case of DNNs with ReLU, for the purpose of analysis, we can replace the hard-ReLU gate by the soft-ReLU gate. Now, the gradient is given by $\partial_\theta \hat{y}_\Theta(x) = \langle \partial_\theta \phi_{x,\Theta}, v_\Theta \rangle + \langle \phi_{x,\Theta}, \partial_\theta v_\Theta \rangle$, where the two terms on the right can be interpreted as NPF and NPV gradients. Using the soft-ReLU ensures $\psi^f \neq 0$ (note that $\psi^f = 0$ for hard-ReLU due to its $0/1$ state). We can obtain further insights in the DLNPF regime, wherein, the NTK is given by $K_{\Theta^{DGN}} = K^v_{\Theta^{DGN}} + K^f_{\Theta^{DGN}}$. For MNIST, we compared $K^v_{\Theta^{DGN}}$ and $K^f_{\Theta^{DGN}}$ (calculated on 100 examples in total with 10 examples per each of the 10 classes) using their trace and Frobenius norms, and we observe that $K^v_{\Theta^{DGN}}$ and $K^f_{\Theta^{DGN}}$ are in the same scale (see right plot in Figure 3), which perhaps shows that both $K^f_{\Theta^{DGN}}$ and $K^v_{\Theta^{DGN}}$ are equally important for obtaining good generalisation performance. Studying $K^f_{\Theta^{DGN}}$ responsible for NPF learning is an interesting future research direction.

## 7   Related Work

**Gated Linear Unit (GaLU)** networks with a single hidden layer was theoretically analysed by Fiat et al. [2019a,b]. In contrast, our NPFs/NPV formulation enabled us to analyse DGNs of any depth $d$. The fixed random filter setting of Fiat et al. [2019b] is equivalent to (in our setting) the FRNPF regime of DGNs with a single hidden layer. They test the hypothesis (on MNIST and Fashion-MNIST) that the effectiveness of ReLU networks is mainly due to the training of the linear part (i.e., weights) and not the gates. They also show that a ReLU network marginally outperforms a GaLU network (both networks have a single hidden layer), and propose a non-gradient based algorithm (which has a separate loop to randomly search and replace the gates of the GaLU network) to close the margin between the GaLU and ReLU networks. We observe a similar trend in that, the FRNPFs (here only NPV is learnt) do perform well in the experiments with a test accuracy of around $68\%$ on CIFAR-10. However, it is also the case that models with NPF learning perform significantly better (by more than $10\%$) than FRNPF, which underscores the importance of the gates. Further, we used soft-ReLU gates in the DLNPF regime, and showed that using standard optimisers (based on gradient descent), we can learn the NPFs and NPV in two separate networks. In addition, capturing the role of the active sub-networks via the NPK is another significant progress over Fiat et al. [2019a,b].

**Prior NTK works:** Jacot et al. [2018] showed the NTK to be the central quantity in the study of generalisation properties of infinite width DNNs. Jacot et al. [2019] identify two regimes that occur at initialisation in fully connected infinite width DNNs namely i) *freeze:* here, the (scaled) NTK converges to a constant and hence leads to slow training, and ii) *chaos:* here, the NTK converges to Kronecker delta and hence hurts generalisation. Jacot et al. [2019] also suggest that for good generalisation it is important to operate the DNNs at the edge of the freeze and the chaos regimes. The works of Arora et al. [2019a], Cao and Gu [2019] are closely related to our work and have been discussed in the introduction. Du et al. [2018] use the NTK to show that over-parameterised DNNs trained by gradient descent achieve zero training error. Du and Hu [2019], Shamir [2019], Saxe et al. [2013] studied deep linear networks (DLNs). Since DLNs are special cases of DGNs, Theorem 5.1 of our paper also provides an expression for the NTK at initialisation of deep linear networks. To see this, in the case of DLNs, all the gates are always $1$ and $\Lambda_\Theta$ is a matrix whose entries will be $w^{(d-1)}$.

**Other works:** Empirical analysis of the role of the gates was done by Srivastava et al. [2014], where the active sub-networks are called as *locally competitive* networks. Here, a 'sub-mask' encodes the $0/1$ state of all the gates. A 't-SNE' visualisation of the sub-masks showed that "subnetworks active for examples of the same class are much more similar to each other compared to the ones activated for the examples of different classes". Balestriero et al. [2018] connect max-affine linearity and DNN with ReLU activations. Neyshabur et al. [2015] proposed a novel *path-norm* based gradient descent.

## 8   Conclusion

In this paper, we developed a novel neural path framework to capture the role of gates in deep learning. We showed that the neural path features are learnt during training and such learning improves generalisation. In our experiments, we observed that almost all information of a trained DNN is stored in the neural path features. We conclude by saying that *understanding deep learning requires understanding neural path feature learning*.

## 9 Broader Impact

Deep neural networks are still widely regarded as blackboxes. The standard and accepted view on the inner workings of deep neural networks is the 'layer-by-layer' viewpoint: as the input progresses through the hidden layers, features at different levels of abstractions are learnt. This paper deviates from the standard 'layer-by-layer' viewpoint, in that, it breaks down the deep neural network blackbox into its constituent paths: different set of paths get fired for different inputs, and the output is the summation of the contribution from individual paths. This makes the inner workings of deep neural networks interpretable, i.e., each input is remembered in terms of the active sub-network of the paths that get 'fired' for that input, and learning via gradient descent amounts to 'rewiring' of the paths. The paper also analytically connects this sub-network and path based view to the recent kernel based interpretation of deep neural networks, and furthers the understanding of feature learning in deep neural networks. We believe that these better insights into the working of DNNs can potentially lead to foundational algorithmic development in the future.

## Acknowledgements

We thank Harish Guruprasad Ramaswamy, Arun Rajkumar, Prabuchandran K J, Braghadeesh Lakshminarayanan and the anonymous reviewers for their valuable comments. We would also like to thank Indian Institute of Technology Palakkad for the 'Seed Grant', and Science and Engineering Research Board (SERB), Department of Science and Technology, Government of India for the 'Startup Research Grant' (SRG/2019/001856).

## Footnotes

[2]$a_t = \mathcal{O}(b_t)$ if $\limsup_{t \to \infty} |a_t / b_t| < \infty$, and $\tilde{\mathcal{O}}(\cdot)$ is used to hide logarithmic factors in $\mathcal{O}(\cdot)$.

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
