[Supplementary Material]

# Appendix

## A  Expression for $K^{(d)}$

The $K^{(d)}$ matrix is computed by the recursion in (2).

$$\tilde{K}^{(1)}(s, s') = \Sigma^{(1)}(s, s') = \Sigma(s, s'), M_{ss'}^{(l)} = \begin{bmatrix} \Sigma^{(l)}(s, s) & \Sigma^{(l)}(s, s') \\ \Sigma^{(l)}(s', s) & \Sigma^{(l)}(s', s') \end{bmatrix} \in \mathbb{R}^2,$$

$$\Sigma^{(l+1)}(s, s') = 2 \cdot \mathbb{E}_{(q,q') \sim N(0, M_{ss'}^{(l)})} [\chi(q)\chi(q')], \hat{\Sigma}^{(l+1)}(s, s') = 2 \cdot \mathbb{E}_{(q,q') \sim N(0, M_{ss'}^{(l)})} [\partial\chi(q)\partial\chi(q')],$$

$$\tilde{K}^{(l+1)} = \tilde{K}^{(l)} \odot \hat{\Sigma}^{(l+1)} + \Sigma^{(l+1)}, K^{(d)} = \left( \tilde{K}^{(d)} + \Sigma^{(d)} \right)/2 \tag{2}$$

where $s, s' \in [n]$ are two input examples in the dataset, $\Sigma$ is the data Gram matrix, $\partial\chi$ stands for the derivative of the activation function with respect to the pre-activation input, $N(0, M)$ stands for the mean-zero Gaussian distribution with co-variance matrix $M$.

## B  Proofs of technical results

**Proof of Proposition 1.1**

*Proof.* We know that $e_t = (e_t(s), s \in [n]) \in \mathbb{R}^n$, and $e_t(s) = \hat{y}_{\Theta_t}(x_s) - y(s)$. Now

$$L_{\Theta_t} = \frac{1}{2} \sum_{s'=1}^{n} (\hat{y}_{\Theta_t} - y)^2$$

$$= \frac{1}{2} \sum_{s'=1}^{n} e_t^2$$

$$\nabla_\Theta L_{\Theta_t} = \sum_{s'=1}^{n} \nabla_\Theta \hat{y}_{\Theta_t}(x_{s'}) e_t(s')$$

$$\nabla_\Theta L_{\Theta_t} = \sum_{s'=1}^{n} \psi_{x_{s'}, \Theta_t} e_t(s') \tag{3}$$

For gradient descent, $\dot{\Theta}_t = -\nabla_\Theta L_{\Theta_t}$, from (3) it follows that

$$\dot{\Theta}_t = -\sum_{s'=1}^{n} \psi_{x_{s'}, \Theta_t} e_t(s') \tag{4}$$

Now $\dot{e}_t = \dot{\hat{y}}_{\Theta_t}$, and expanding $\dot{\hat{y}}_{\Theta_t}(x_s)$ for some $s \in [n]$, we have:

$$\dot{\hat{y}}_{\Theta_t}(x_s) = \frac{d\hat{y}_{\Theta_t}(x_s)}{dt}$$

$$= \sum_{\theta \in \Theta} \frac{d\hat{y}_{\Theta_t}(x_s)}{d\theta} \frac{d\theta_t}{dt}, \text{ by expressing this summation as a dot product we obtain}$$

$$\dot{\hat{y}}_{\Theta_t}(x_s) = \langle \psi_{x_s, \Theta_t}, \dot{\Theta}_t \rangle \tag{5}$$

We now use that fact that $\Theta_t$ is updated by gradient descent

$$\dot{\hat{y}}_{\Theta_t}(x_s) = -\langle \psi_{x_s, \Theta_t}, \sum_{s'=1}^{n} \psi_{x_{s'}, \Theta_t} e_t(s') \rangle$$

$$= -\sum_{s'=1}^{n} K_{\Theta_t}(s, s') e_t(s') \tag{6}$$

The proof is complete by recalling that $\hat{y}_{\Theta_t} = (\hat{y}_{\Theta_t}(x_s), s \in [n])$, and $\dot{e}_t = \dot{\hat{y}}_{\Theta_t}$. □

**Proof of Proposition 2.1**

*Proof.* Let $x \in \mathbb{R}^{d_{\text{in}}}$ be the input to the DNN and $\hat{y}_\Theta(x)$ be its output. The output can be written in terms of the final hidden layer output as:

$$\hat{y}_\Theta(x) = \sum_{j_{d-1}=1}^{w} \Theta(1, j_{d-1}, d) \cdot z_{x,\Theta}(j_{d-1}, d-1)$$

$$= \sum_{j_{d-1}=1}^{w} \Theta(1, j_{d-1}, d) \cdot G_{x\Theta}(j_{d-1}, d-1) \cdot q_{x,\Theta}(j_{d-1}, d-1) \quad (7)$$

Now $q_{x,\Theta}(j_{d-1}, d-1)$ for a fixed $j_{d-1}$ can again be expanded as

$$q_{x,\Theta}(j_{d-1}, d-1) = \sum_{j_{d-2}=1}^{w} \Theta(j_{d-1}, j_{d-2}, d-1) \cdot z_{x,\Theta}(j_{d-2}, d-2)$$

$$= \sum_{j_{d-2}=1}^{w} \Theta(j_{d-1}, j_{d-2}, d-1) \cdot G_{x,\Theta}(j_{d-2}, d-2) \cdot q_{x,\Theta}(j_{d-2}, d-2) \quad (8)$$

Now plugging in (8) in the expression in (7), we have

$$\hat{y}_\Theta(x) = \sum_{j_{d-1}=1}^{w} \Theta(1, j_{d-1}, d) \cdot G_{x\Theta}(j_{d-1}, d-1) \left( \sum_{j_{d-2}=1}^{w} \Theta(j_{d-1}, j_{d-2}, d-1) \right.$$

$$\left. \cdot G_{x,\Theta}(j_{d-2}, d-2) \cdot q_{x,\Theta}(j_{d-2}, d-2) \right)$$

$$= \sum_{j_{d-1}, j_{d-2} \in [w]} G_{x,\Theta}(j_{d-1}, d-1) \cdot G_{x,\Theta}(j_{d-2}, d-2) \cdot \Theta(1, j_{d-1}, d)$$

$$\cdot \Theta(j_{d-1}, j_{d-2}, d-1) \cdot q_{x,\Theta}(j_{d-2}, d-2)$$

$$(9)$$

By expanding $q$'s for all the previous layers till the input layer we have

$$\hat{y}_\Theta(x) = \sum_{j_d=1, j_{d-1}, \ldots, j_1 \in [w], j \in [d_{\text{in}}]} x(j) \Pi_{l=1}^{d-1} G_{x,\Theta}(j_l, l) \Pi_{l=1}^{d} \Theta(j_l, j_{l-1}, l)$$

$\square$

**Proof of Lemma 2.1**

*Proof.*

$$\langle \phi_{x_s,\Theta}, \phi_{x_{s'},\Theta} \rangle = \sum_{p \in [P]} x_s(\mathcal{I}_0(p)) x_{s'}(\mathcal{I}_0(p)) A_\Theta(x_s, p) A_\Theta(x_{s'}, p)$$

$$= \sum_{i=1}^{d_{\text{in}}} x_s(i) x_{s'}(i) \Lambda_\Theta(s, s')$$

$$= \langle x_s, x_{s'} \rangle \cdot \Lambda_\Theta(s, s') \quad (10)$$

$\square$

**Proof of Proposition 3.1**

*Proof.* Let $\Psi_\Theta = (\psi_{x_s,\Theta}, s \in [n]) \in \mathbb{R}^{d_{\text{net}} \times n}$ be the NTF matrix, then the NTK matrix is given by $K_{\Theta_t} = \Psi_{\Theta_t}^\top \Psi_{\Theta_t}$. Note that, $\hat{y}_\Theta(x_s) = \langle \phi_{x_s,\Theta}, v_\Theta \rangle = \langle v_\Theta, \phi_{x_s,\Theta} \rangle = v_\Theta^\top \phi_{x_s,\Theta}$. Now $\psi_{x_s,\Theta} = \nabla_\Theta v_\Theta \phi_{x_s,\Theta}$, and hence $\Psi = \nabla_\Theta v_\Theta \Phi_\Theta$. Hence, $K_{\Theta_t} = \Psi_{\Theta_t}^\top \Psi_{\Theta_t} = \Phi_\Theta^\top (\nabla_\Theta v_\Theta)^\top (\nabla_\Theta v_\Theta) \Phi_\Theta = \Phi_\Theta^\top \mathcal{V}_\Theta \Phi_\Theta$. $\square$

**Proof of Proposition 3.2**

*Proof.* Follows in a similar manner as the proof of Proposition 1.1. □

**Proof of Proposition 3.3**

*Proof.* $\rho_{\min}(K_\Theta) = \min\limits_{\substack{x \in \mathbb{R}^n \\ \|x\|_2 = 1}} x^\top K_\Theta x$. Let $x' \in \mathbb{R}^n$ such that $\|x'\|_2 = 1$ and $\rho_{\min}(H_\Theta) = {x'}^\top H_\Theta x'$.

Now, $\rho_{\min}(K_\Theta) \leq {x'}^\top K_\Theta x'$. Let $y' = \Phi x'$, then we have, $\rho_{\min}(K_\Theta) \leq {y'}^\top \mathcal{V}_\Theta y'$. Hence $\rho_{\min}(K_\Theta) \leq \|y'\|_2^2 \rho_{\max}(\mathcal{V}_\Theta)$. Proof is complete by noting that $\|y'\|_2^2 = {x'}^\top \Phi_\Theta^\top \Phi_\Theta x' = \rho_{\min}(H_\Theta)$. □

**Proof of Proposition 4.1**

*Proof.* Follows in a similar manner as proof of Proposition 1.1. □

## B.1 Proof of Theorem 5.1

### B.1.1 Calculation of $\mathbb{E}\left[K^{\mathbf{v}}_{\Theta_0^{\mathbf{DGN}}}\right]$

**Proposition B.1.** *Let $\theta^v \in \Theta^v$ be a weight in layer $l_{\theta^v}$, and let $p$ be a path that passes through $\theta^v$. Then*

$$\partial_{\theta^v} v_{\Theta^v}(p) = \Pi_{l=1, l \neq l_{\theta^v}}^d \Theta(\mathcal{I}_l(p), \mathcal{I}_{l-1}(p), l) \tag{11}$$

*Proof.* Proof follows by noting that $v_{\Theta^v}(p) = \Pi_{l=1}^d \Theta(\mathcal{I}_l(p), \mathcal{I}_{l-1}(p), l)$. □

**Lemma B.1.** *Let $\varphi_{p,\Theta}$ be as in Definition 3.1, under the assumption in Theorem 5.1, for paths $p_1, p_2 \in [P], p_1 \neq p_2$, at initialisation we have (i) $\mathbb{E}\left[\langle \varphi_{p_1,\Theta_0^v}, \varphi_{p_2,\Theta_0^v}\rangle\right] = 0$, (ii) $\langle \varphi_{p_1,\Theta_0^v}, \varphi_{p_1,\Theta_0^v}\rangle = d \cdot \sigma^{2(d-1)}$.*

*Proof.*

$$\langle \varphi_{p_1,\Theta_0^v}, \varphi_{p_2,\Theta_0^v}\rangle = \sum_{\theta^v \in \Theta^v} \partial_{\theta^v} v_{\Theta_0^v}(p_1) \partial_{\theta^v} v_{\Theta_0^v}(p_2)$$

Let $\theta^v \in \Theta^v$ be an arbitrary weight. If either $p_1$ or $p_2$ does not pass through $\theta^v$, then it follows that $\partial_{\theta^v} v_{\Theta_0^v}(p_1) \partial_{\theta^v} v_{\Theta_0^v}(p_2) = 0$. Let us consider the the case when $p_1, p_2$ pass through $\theta^v$ and without of loss of generality let $\theta^v$ belong to layer $l_{\theta^v} \in [d]$. we have

$$\mathbb{E}\left[\partial_{\theta^v} v_{\Theta_0^v}(p_1) \partial_{\theta^v} v_{\Theta_0^v}(p_2)\right]$$

$$= \mathbb{E}\left[\prod_{\substack{l=1 \\ l \neq l_{\theta^v}}}^d \left(\Theta_0^v(\mathcal{I}_l(p_1), \mathcal{I}_{l-1}(p_1), l) \Theta_0^v(\mathcal{I}_l(p_2), \mathcal{I}_{l-1}(p_2), l)\right)\right]$$

$$= \prod_{\substack{l=1 \\ l \neq l_{\theta^v}}}^d \mathbb{E}\left[\Theta_0^v(\mathcal{I}_l(p_1), \mathcal{I}_{l-1}(p_1), l) \Theta_0^v(\mathcal{I}_l(p_2), \mathcal{I}_{l-1}(p_2), l)\right]$$

where the $\mathbb{E}[\cdot]$ moved inside the product because at initialisation the weights (of different layers) are independent of each other. Since $p_1 \neq p_2$, there exist a layer $\tilde{l} \in [d], \tilde{l} \neq l_{\theta^v}$ such that they do not pass through the same weight in layer $\tilde{l}$, i.e., $\Theta_0^v(\mathcal{I}_{\tilde{l}}(p_1), \mathcal{I}_{\tilde{l}-1}(p_1), \tilde{l},)$ and $\Theta_0^v(\mathcal{I}_{\tilde{l}}(p_2), \mathcal{I}_{\tilde{l}-1}(p_2), \tilde{l})$

are distinct weights. Using this fact, we have

$$\mathbb{E}\left[\partial_{\theta^v}v_{\Theta_0^v}(p_1)\partial_{\theta^v}v_{\Theta_0^v}(p_2)\right]$$

$$=\left(\prod_{\substack{l=1\\l\neq l_{\theta^v},\tilde{l}}}^{d}\mathbb{E}\left[\Theta_0^v(\mathcal{I}_l(p_1),\mathcal{I}_{l-1}(p_1),l)\Theta_0^v(\mathcal{I}_l(p_2),\mathcal{I}_{l-1}(p_2),l)\right]\right)$$

$$\cdot\left(\mathbb{E}\left[\Theta_0^v(\mathcal{I}_{\tilde{l}}(p_1),\mathcal{I}_{\tilde{l}-1}(p_1),\tilde{l})\right]\mathbb{E}\left[\Theta_0^v(\mathcal{I}_{\tilde{l}}(p_2),\mathcal{I}_{\tilde{l}-1}(p_2),\tilde{l})\right]\right)$$

$$=0$$

The proof of (ii) is complete by noting that a given path $p_1$ pass through only '$d$' weights, and hence $\sum_{\theta^v\in\Theta^v}\partial_{\theta^v}v_{\Theta_0^v}(p_1)\partial_{\theta^v}v_{\Theta_0^v}(p_1)$ has '$d$' non-zero terms, and the fact that at initialisation we have

$$\partial_{\theta^v}v_{\Theta_0^v}(p_1)\partial_{\theta^v}v_{\Theta_0^v}(p_1)$$

$$=\prod_{\substack{l=1\\l\neq l_{\theta^v}}}^{d}[\Theta_0^v(\mathcal{I}_l(p),\mathcal{I}_{l-1}(p),l)]^2$$

$$=\sigma^{2(d-1)}$$

$\square$

**Theorem B.1.** $\mathbb{E}\left[K_{\Theta_0^{DGN}}^v\right]=d\cdot\sigma^{2(d-1)}\cdot H_{FNPF}$.

*Proof.* Let $\Phi_{\text{FNPF}}=\Phi_{\Theta_0^f}=\left(\phi_{x_s,\Theta_0^f},s\in[n]\right)\in\mathbb{R}^{P\times n}$ be the NPF matrix.

$$\mathbb{E}\left[K_{\Theta_0^{DGN}}^v\right]=\mathbb{E}\left[\Phi_{\text{FNPF}}^\top\mathcal{V}_{\Theta_0^v}\Phi_{\text{FNPF}}\right]$$

$$=\mathbb{E}\left[\Phi_{\text{FNPF}}^\top(\nabla_{\Theta^v}v_{\Theta_0^v})^\top(\nabla_{\Theta^v}v_{\Theta_0^v})\Phi_{\text{FNPF}}\right]$$

$$=\Phi_{\text{FNPF}}^\top\left(\mathbb{E}\left[(\nabla_{\Theta^v}v_{\Theta_0^v})^\top(\nabla_{\Theta^v}v_{\Theta_0^v})\right]\right)\Phi_{\text{FNPF}}$$

$$\overset{(a)}{=}d\cdot\sigma^{2(d-1)}\cdot\left(\Phi_{\text{FNPF}}^\top\Phi_{\text{FNPF}}\right)$$

$$=d\cdot\sigma^{2(d-1)}\cdot H_{\text{FNPF}}$$

Here, $(a)$ follows from Lemma B.1, i.e., $\mathbb{E}\left[(\nabla_{\Theta^v}v_{\Theta_0^v})^\top(\nabla_{\Theta^v}v_{\Theta_0^v})\right]=d\cdot\sigma^{2(d-1)}\cdot I_{P\times P}$, where $I_{P\times P}$ is a $P\times P$ identity matrix. $\square$

### B.1.2 Calculation of $Var\left[K_{\Theta_0^{DGN}}^v\right]$

**Notation:** For $x,x'\in\mathbb{R}^{d_{\text{in}}}$, let $\phi(p)=\phi_{x,\Theta_0^f}(p)$, and $\phi'(p)=\phi_{x',\Theta_0^f}(p)$. Also in what follows we use $\theta_a,\theta_b$ to denote the individual weights in the value network, and $p_a,p_a',p_b,p_b'\in[P]$ to denote the paths. Further, unless otherwise specified, quantities $\theta_a,\theta_b,p_a,p_a',p_b,p_b'$ are unrestricted.

**Proposition B.2.**
$$K_{\Theta_0^{DGN}}^v(x,x')=\sum_{\theta_a,p_a,p_a'}\phi(p_a)\phi'(p_a')\partial_{\theta_a}v_{\Theta_0^v}(p_a)\partial_{\theta_a}v_{\Theta_0^v}(p_a') \tag{12}$$

$$\tag{13}$$

*Proof.*
$$K_{\Theta_0^{DGN}}^v(x,x')=\langle\nabla_{\Theta^v}\hat{y}_{\Theta_0^{DGN}}(x),\nabla_{\Theta^v}\hat{y}_{\Theta_0^{DGN}}(x')\rangle \tag{14}$$

$$=\sum_{\theta_a\in\Theta^v}\left(\sum_{p_a\in[P]}\phi(p_a)\partial_{\theta_a}v_{\Theta_0^v}(p_a)\right)\left(\sum_{p_a'\in[P]}\phi'(p_a')\partial_{\theta_a}v_{\Theta_0^v}(p_a')\right) \tag{15}$$

$$=\sum_{\theta_a,p_a,p_a'}\phi(p_a)\phi'(p_a')\partial_{\theta_a}v_{\Theta_0^v}(p_a)\partial_{\theta_a}v_{\Theta_0^v}(p_a') \tag{16}$$

$\square$

We now drop $\Theta^v$ in $v_{\Theta_0^v}$ and v, $\Theta_0^{\text{DGN}}$ from $K_{\Theta_0^{\text{DGN}}}^v$, and we denote $\Theta^v$ by $\Theta$.

**Proposition B.3.**

$$\mathbb{E}\left[K(x, x')\right] = \sum_{\theta_a, p_a} \phi(p_a)\phi'(p_a)\mathbb{E}\left[\left(\partial_{\theta_a} v(p_a)\right)^2\right] \tag{17}$$

$$\mathbb{E}\left[K^2(x, x')\right] = \sum_{\substack{\theta_a, p_a, p'_a \\ \theta_b, p_b, p'_b}} \phi(p_a)\phi'(p'_a)\phi(p_b)\phi'(p'_b)\mathbb{E}\left[\partial_{\theta_a} v(p_a)\partial_{\theta_a} v(p'_a)\partial_{\theta_b} v(p_b)\partial_{\theta_b} v(p'_b)\right] \tag{18}$$

*Proof.*

$$\mathbb{E}\left[K(x, x')\right] = \sum_{\theta_a, p_a, p'_a} \phi(p_a)\phi'(p'_a)\mathbb{E}\left[\partial_{\theta_a} v(p_a)\partial_{\theta_a} v(p'_a)\right] \tag{19}$$

$$\overset{(a)}{=} \sum_{\theta_a, p_a} \phi(p_a)\phi'(p_a)\mathbb{E}\left[\left(\partial_{\theta_a} v(p_a)\right)^2\right] \tag{20}$$

where $(a)$ follows from Lemma B.1 that for $p_a \neq p'_a$ $\mathbb{E}\left[\partial_{\theta_a} v(p_a)\partial_{\theta_a} v(p'_a)\right] = 0$.

The expression for $\mathbb{E}\left[K^2(x, x')\right]$ is obtained by squaring the expression in (12) and pushing the $\mathbb{E}\left[\cdot\right]$ inside the summation.

$\square$

**Definition B.1.**

1. *Let $\tau = (p_a, p'_a, p_b, p'_b; \theta_a, \theta_b)$ denote the index used to sum the terms in the expression for $\mathbb{E}\left[K^2(x, x')\right]$ given in (18). Note that the index contains 4 path variables namely $p_a, p'_a, p_b, p'_b$ and 2 weight variables namely $\theta_a, \theta_b$.*

2. *An index $\tau = (p_a, p'_a, p_b, p'_b; \theta_a, \theta_b)$ is said to correspond to a 'base' term if $p_a = p'_a$ and $p_b = p'_b$. We define $\mathcal{B}$ to be the set of indices corresponding to 'base' terms given by $\mathcal{B} = \{\tau = (p_a, p'_a, p_b, p'_b; \theta_a, \theta_b) : p_a = p'_a, p_b = p'_b\}$.*

3. *For $\tau = (p_a, p'_a, p_b, p'_b; \theta_a, \theta_b)$, define $\omega(\tau) \overset{def}{=} \phi(p_a)\phi'(p'_a)\phi(p_b)\phi'(p'_b)$.*

4. *For $\tau = (p_a, p'_a, p_b, p'_b; \theta_a, \theta_b)$, define $E(\tau) \overset{def}{=} \mathbb{E}\left[\partial_{\theta_a} v(p_a)\partial_{\theta_a} v(p'_a)\partial_{\theta_b} v(p_b)\partial_{\theta_b} v(p'_b)\right]$.*

**Remark:** Definition B.1 helps us to re-write (18) as $\mathbb{E}\left[K^2(x, x')\right] = \sum_\tau \omega(\tau)E(\tau)$.

**Proposition B.4.** *For $\tau = (p_a, p'_a, p_b, p'_b; \theta_a, \theta_b)$, let $\theta_a$ belong to layer $l_a \in [d]$ and $\theta_b$ belong to layer $l_b \in [d]$. Let paths $p_a$ and $p'_a$ pass through $\theta_a$ and paths $p_b$ and $p'_b$ pass through $\theta_b$.*

$$E(\tau) = \underbrace{\Pi_{\substack{l=1 \\ l \neq l_b \\ l \neq l_a}}^d \mathbb{E}\left[\Theta(\mathcal{I}_l(p_a), \mathcal{I}_{l-1}(p_a), l)\Theta(\mathcal{I}_l(p'_a), \mathcal{I}_{l-1}(p'_a), l)\Theta(\mathcal{I}_l(p_b), \mathcal{I}_{l-1}(p_b), l)\Theta(\mathcal{I}_l(p'_b), \mathcal{I}_{l-1}(p'_b), l)\right]}_{\text{Term-I}}$$

$$\cdot \underbrace{\mathbb{E}\left[\Theta(\mathcal{I}_{l_b}(p_a), \mathcal{I}_{l_b-1}(p_a), l_b)\Theta(\mathcal{I}_{l_b}(p'_a), \mathcal{I}_{l_b-1}(p'_a), l_b)\right]}_{\text{Term-II}}$$

$$\cdot \underbrace{\mathbb{E}\left[\Theta(\mathcal{I}_{l_a}(p_b), \mathcal{I}_{l_a-1}(p_b), l_a)\Theta(\mathcal{I}_{l_a}(p'_b), \mathcal{I}_{l_a-1}(p'_b), l_a)\right]}_{\text{Term-III}} \tag{21}$$

*Proof.* Since the paths $p_a, p'_a$ pass through $\theta_a$ and paths $p_b, p'_b$ pass through $\theta_b$, it follows that $\partial_{\theta_a} v(p_a) \neq 0$, $\partial_{\theta_a} v(p'_a) \neq 0$, $\partial_{\theta_b}(p_b) \neq 0$ and $\partial_{\theta_b}(p'_b) \neq 0$. Note that,

$$\partial_{\theta_a} v(p_a) = \Pi_{l=1, l\neq l_a}^d \Theta(\mathcal{I}_l(p_a), \mathcal{I}_{l-1}(p_a), l)$$

$$\partial_{\theta_a} v(p'_a) = \Pi_{l=1, l\neq l_a}^d \Theta(\mathcal{I}_l(p'_a), \mathcal{I}_{l-1}(p'_a), l)$$

$$\partial_{\theta_a} v(p_b) = \Pi_{l=1, l\neq l_b}^d \Theta(\mathcal{I}_l(p_b), \mathcal{I}_{l-1}(p_b), l)$$

$$\partial_{\theta_a} v(p'_b) = \Pi_{l=1, l\neq l_b}^d \Theta(\mathcal{I}_l(p'_b), \mathcal{I}_{l-1}(p'_b), l)$$

The proof is complete by using the fact that weights of different layers are independent and pushing the $\mathbb{E}$ operator inside the $\mathbb{E}\left[\partial_{\theta_a}v(p_a)\partial_{\theta_a}v(p'_a)\partial_{\theta_b}v(p_b)\partial_{\theta_b}v(p'_b)\right]$ to convert the expectation of products into a product of expectations. $\qquad\square$

**Proposition B.5.** *For $\tau = (p_a, p'_a, p_b, p'_b; \theta_a, \theta_b)$, $E(\tau) = \sigma^{4(d-1)}$ if and only if*

• *Condition I: $p_a, p'_a$ pass through $\theta_a$ and $p_b, p'_b$ pass through $\theta_b$.*

• *Condition II: In every layer, $l \in [d]$ either all the 4 paths $p_a, p'_a, p_b, p'_b$ pass through the same weight or there exists two distinct weights, say $\theta_{I,l}$ and $\theta_{II,l}$ such that, 2 paths out of $p_a, p'_a, p_b, p'_b$ pass through $\theta_{I,l}$ and the other 2 paths pass through $\theta_{II,l}$.*

*Proof.*

**Sufficiency:** If **Condition I** and **Condition II** hold, then from (21) it follows that $E(\tau) = \sigma^{4(d-2)} \cdot \sigma^2 \cdot \sigma^2 = \sigma^{4(d-1)}$.

**Necessity:** If **Condition I** does not hold, then either one of $\partial_{\theta_a}v(p_a), \partial_{\theta_a}v(p'_a), \partial_{\theta_b}v(p_b), \partial_{\theta_b}v(p'_b)$ becomes 0. If **Condition II** does not hold, either Term-I or Term-II or Term-III in (21) evaluates to 0 because all the weights involved are independent symmetric Bernoulli. $\qquad\square$

**Definition B.2** (Crossing). *Paths $\rho_a$ and $\rho_b$ are said to cross each other if they pass through the same node in one or one or more of the intermediate layers $l = 2, \ldots, d-1$. For the sake of consistency, for paths $\rho_a$ and $\rho_b$ that do not cross, we call them to have 0 crossings.*

**Definition B.3** (Splicing). *Let $(\rho_a, \rho_a, \rho_b, \rho_b)$ be 4 paths (from a base term) occurring in pairs of 2 each. Let $\rho_a$ and $\rho_b$ cross at $k \in \{0, \ldots, d-1\}$ intermediate nodes, belonging to layers $l_1, \ldots, l_k$ (let $l_0 = 0$ and $l_{k+1} = d$). Let the set of permutations of $(a, a, b, b)$ be denoted by $Pm\left((a, a, b, b)\right) \subset \{a, b\}^4$. We say that paths $(p_a, p'_a, p_b, p'_b)$ to be 'splicing' of $(\rho_a, \rho_a, \rho_b, \rho_b)$ if there exists $base(i, \cdot) \in Pm\left((a, a, b, b)\right), i = 1, \ldots, k+1$ such that*

$$I_l(p_a) = I_l(\rho_{base(i,1)}), l \in [l_{i-1}, l_i], i = 1, \ldots, k+1$$
$$I_l(p'_a) = I_l(\rho_{base(i,2)}), l \in [l_{i-1}, l_i], i = 1, \ldots, k+1$$
$$I_l(p_b) = I_l(\rho_{base(i,3)}), l \in [l_{i-1}, l_i], i = 1, \ldots, k+1$$
$$I_l(p'_b) = I_l(\rho_{base(i,4)}), l \in [l_{i-1}, l_i], i = 1, \ldots, k+1$$

**Lemma B.2.** *Let $\tau = (p_a, p'_a, p_b, p'_b; \theta_a, \theta_b)$ be such that $E(\tau) = \sigma^{4(d-1)}$. Then there exists $\rho_a$ and $\rho_b$ such that $\rho_a$ passes through $\theta_a$, and $\rho_b$ passes through $\theta_b$, and $(p_a, p'_a, p_b, p'_b)$ is a splicing of $(\rho_a, \rho_a, \rho_b, \rho_b)$.*

*Proof.* Using Proposition B.5 and the fact that $p_a, p'_a, p_b, p'_b$ are paths, only the layouts shown in Figure 4 are possible. In Figure 4, the 4 different coloured lines stand for the 4 different paths namely $p_a, p'_a, p_b, p'_b$. The hidden nodes are denoted by the circles. Here, **(a)** is the case where all the 4 paths pass through the same weight in a given layer. **(b),(c), (d)** are the cases where 2 paths out of the 4 paths pass through one weight and the other 2 paths pass through a different weight in a given layer. Table 3 provides the conditions for the possible current and next layer layouts.

Figure 4: Various ways in which the 4 paths $p_a, p'_a, p_b, p'_b$ can pass through a given layer.

| Current Layer Layout | Next Layer Layout |
|:---:|:---:|
| **(a)** | **(a)** or **(b)** |
| **(b)** | **(c)** or **(d)** |
| **(c)** | **(a)** or **(b)** |
| **(d)** | **(c)** or **(d)** |

Table 3: Show the possible current and next layer layouts.

Thus in each layer $p_a, p_a', p_b, p_b'$ can always be paired to obtain $\rho_a$ and $\rho_b$. In the splicing, $base(i, 1)$ specifies whether $p_a$ follows $\rho_a$ or $\rho_b$ between layers $l_{i-1}$ and $l_i$ (i.e., between crossing points). The role of $base(i, 2)$, $base(i, 3)$ and $base(i, 4)$ can be explained in a similar manner. $\qquad\square$

**Lemma B.3.** *Let $\tau' = (\rho_a, \rho_a, \rho_b, \rho_b; \theta_a, \theta_b) \in \mathcal{B}$ be an index in the base set such that $\rho_a$ and $\rho_b$ do not cross and $E(\tau') \neq 0$. Let $\tau = (p_a, p_a', p_b, p_b'; \theta_a, \theta_b)$ be such that $(p_a, p_a', p_b', p_b) \neq (\rho_a, \rho_a, \rho_b, \rho_b)$ is a 'splicing' of $(\rho_a, \rho_a, \rho_b, \rho_b)$. Then $E(\tau) = 0$.*

*Proof.* Since $E(\tau') \neq 0$, it follows that $\rho_a$ passes through $\theta_a$ and $\rho_b$ passes through $\theta_b$. Since $\rho_a$ and $\rho_b$ do not cross each other, the only possible splicings are the permutations of $(\rho_a, \rho_a, \rho_b, \rho_b)$ itself. For the sake of concreteness, let us pick a $\tau$ such that $(p_a, p_a', p_b', p_b) = (\rho_a, \rho_b, \rho_a, \rho_b)$ (a non-identity permutation). For $E(\tau) \neq 0$ to hold, $\partial_{\theta_a} v(\rho_b) \neq 0$ and $\partial_{\theta_b} v(\rho_a) \neq 0$ should also hold, which implies both $\rho_a$ and $\rho_b$ pass through $\theta_a$ and $\theta_b$. However, we assumed that $\rho_a$ and $\rho_b$ do not cross each other. Hence, $E(\tau) = 0$ for any $\tau$ such that $(p_a, p_a', p_b', p_b)$ is a non-identity permutation of $(\rho_a, \rho_a, \rho_b, \rho_b)$. $\qquad\square$

**Proposition B.6.** *Let $\tau$ and $\mathcal{B}$ be as in Definition B.1, then*

$$\mathbb{E}\left[K(x, x')\right]^2 = \sum_{\tau \in \mathcal{B}} \omega(\tau) E(\tau)$$

*Proof.* Writing down the left-hand and right-hand sides, we have:

$$\mathbb{E}\left[K(x, x')\right]^2 = \sum_{\substack{\theta_a, p_a \\ \theta_b, p_b}} \phi(p_a)\phi'(p_a)\phi(p_b)\phi'(p_b)\mathbb{E}\left[(\partial_{\theta_a} v(p_a))^2\right]\mathbb{E}\left[(\partial_{\theta_b} v(p_b))^2\right]$$

$$\sum_{\tau \in \mathcal{B}} \omega(\tau) E(\tau) = \sum_{\substack{\theta_a, p_a \\ \theta_b, p_b}} \phi(p_a)\phi'(p_a)\phi(p_b)\phi'(p_b)\mathbb{E}\left[(\partial_{\theta_a} v(p_a))^2 (\partial_{\theta_b} v(p_b))^2\right]$$

When $\partial_{\theta_a} v(p_a) \neq 0$ and $\partial_{\theta_b} v(p_b) \neq 0$, for symmetric Bernoulli weights it follows that $\mathbb{E}\left[(\partial_{\theta_a} v(p_a))^2\right]\mathbb{E}\left[(\partial_{\theta_b} v(p_b))^2\right] = \mathbb{E}\left[(\partial_{\theta_a} v(p_a))^2 (\partial_{\theta_b} v(p_b))^2\right] = \sigma^{4(d-1)}$. $\qquad\square$

**Theorem B.2.** *Let the weights be chosen as in Theorem 5.1. Then, it follows that*

$$Var\left[K(x, x')\right] \leq C d_{in}^2 \frac{d^3}{w}$$

*Proof.*

$$
\begin{aligned}
Var\left[K(x, x')\right] &= \mathbb{E}\left[K^2(x, x')\right] - \mathbb{E}\left[K(x, x')\right]^2 \\
&= \sum_{\tau} \omega(\tau) E(\tau) - \sum_{\tau \in \mathcal{B}} \omega(\tau) E(\tau) \\
&= \sum_{\tau \notin \mathcal{B}} \omega(\tau) E(\tau)
\end{aligned}
$$

In what follows, without loss of generality, let $|\omega(\tau)| \leq 1$. Then,

$$Var\left[K(x, x')\right] \leq \sum_{\tau \notin \mathcal{B}} E(\tau)$$

Let $\bar{\mathcal{B}} = \{\tau \notin \mathcal{B}\}$. From Lemma B.2 we know that every $\tau \in \bar{\mathcal{B}}$ such that $E(\tau) = \sigma^{4(d-1)}$ can always be identified with a base term $\tau' = (\rho_a, \rho_a, \rho_b, \rho_b; \theta_a, \theta_b) \in \mathcal{B}$, and from Lemma B.3, we know that in such a $\tau'$, the base paths $\rho_a$ and $\rho_b$ cross $k > 0$ times. Now, there are $\binom{(d-1)}{k} < d^k$ possible ways in which the $k$ crossing can occur within the $(d-1)$ layers, and within each layer there are $w$ possible nodes in which such crossings can occur. The total number of paths that pass through $k < d-1$ nodes is $\frac{P}{w^k}$, where $P = d_{\text{in}} w^{(d-1)}$. And the number of splicings of base terms with $k$ crossings is less than $6^{k+1}$. Once we obtain the paths, the crossings, the splicing, the weights $\theta_a$ and $\theta_b$ can each occur in up to any of the $d-1$ layers. Putting all this together, we have

$$Var\left[K(x, x')\right] \leq \sum_{k=1}^{\infty} d^2 6^{k+1} \cdot (wd)^k \cdot \left(\frac{P^2}{w^{2k}}\right) \sigma^{4(d-1)}$$

$$\leq 6 d_{\text{in}}^2 \sigma'^2 \left(\frac{6d^3}{w}\right) \left(\frac{1}{1 - \frac{6d}{w}}\right)$$

$$\leq C d_{\text{in}}^2 \frac{d^3}{w}$$

$\square$

**Proof of Theorem 5.1**

*Proof.* Follows from Theorem B.1 and Theorem B.2. $\square$

## C  Applying Theorem 5.1 In Finite Width Case

In this section, we describe the technical step in applying Theorem 5.1 which requires $w \to \infty$ to measure the information in the gates of a DNN with finite width. Since we are training only the value network in the FPNP mode of the DGN, it is possible to let the width of the value network alone go to $\infty$, while keeping the width of the feature network (which stores the fixed NPFs) finite. This is easily achieved by multiplying the width by a positive integer $m \in \mathbb{Z}_+$, and *padding* the gates '$m$' times.

**Definition C.1.** *Define $DGN^{(m)}$ to be the DGN whose feature network is of width $w$ and depth $d$, and whose value network is a fully connected network of width $mw$ and depth $d$. The $mw(d-1)$ gating values are obtained by 'padding' the $w(d-1)$ gating values of the width '$w$', depth '$d$' feature network '$m$' times (see Figure 5, Table 4).*

| Feature Network (NPF) | Value Network (NPV) |
|---|---|
| $z_x^{\text{f}}(0) = x$ | $z_x^{\text{v}}(0) = x$ |
| $q_x^{\text{f}}(i, l) = \sum_j \Theta^{\text{f}}(i, j, l) \cdot z_x(j, l-1)$ | $q_x^{\text{v}}(i, l) = \sum_j \Theta^{\text{v}}(i, j, l) \cdot z_x^{\text{v}}(j, l-1)$ |
| $z_x^{\text{f}}(i, l) = q_x^{\text{f}}(i, l) \cdot \mathbb{1}_{\{q_x^{\text{f}}(i,l) > 0\}}$ | $z_x^{\text{v}}(i, l) = q_x^{\text{v}}(i, l) \cdot G_x(i, l)$ |
| None | $\hat{y}_{\Theta^{\text{DGN}(m)}}(x) = \sum_j \Theta^{\text{v}}(1, j, l) \cdot z_x^{\text{v}}(j, d-1)$ |
| Hard ReLU: $G_x(i, l) = \mathbb{1}_{\{q_x^{\text{f}}(i,l) > 0\}}$ or Soft-ReLU: $G_x(i, l) = 1/\left(1 + \exp(-\beta \cdot q_x^{\text{f}}(i, l) > 0)\right)$ | |

Table 4: Deep Gated Network with padding. Here the gating values are padded, i.e., $G_x(kw + i, l) = G_x(i, l), \forall k = 0, 1, \ldots, m-1, i \in [w]$.

**Remark:** $DGN^{(m)}$ has a total of $P^{(m)} = (mw)^{(d-1)} d_{\text{in}}$ paths. Thus, the NPF and NPV are quantities in $\mathbb{R}^{P^{(m)}}$. In what follows, we denote the NPF matrix of $DGN^{(m)}$ by $\Phi_{\Theta_0^{\text{f}}}^{(m)} \in \mathbb{R}^{P^{(m)} \times n}$, and use $H_{\text{FNPF}}^{(m)} = (\Phi_{\Theta_0^{\text{f}}}^{(m)})^{\top} \Phi_{\Theta_0^{\text{f}}}^{(m)}$.

Before we proceed to state the version of Theorem 5.1 for $DGN^{(m)}$, we will look at an equivalent definition for $\Lambda_\Theta$ (see Definition 2.2).

**Definition C.2.** *For input examples $s, s' \in [n]$ define*

*1. $\tau_\Theta(s, s', l) \stackrel{def}{=} \sum_{i=1}^{w} G_{x_s, \Theta}(i, l) G_{x_{s'}, \Theta}(i, l)$ be the number of activations that are "on" for both inputs $s, s' \in [n]$ in layer $l \in [d-1]$.*

2. $\Lambda_{\Theta}(s, s') \overset{def}{=} \Pi_{l=1}^{d-1} \tau_{\Theta}(s, s', l)$.

Figure 5: $\text{DGN}^{(m)}$ where the value network is of width $mw$ and depth $d$. The gates are derived by padding the gating values obtained from the feature network '$m$' times, i.e., $G_x(kw + i, l) = G_x(i, l), \forall k = 0, 1, \ldots, m-1, i \in [w]$.

**Corollary C.1** (Corollary to Theorem 5.1). *Under the same assumptions as in Theorem 5.1 with $\sigma$ replaced by $\sigma_{(m)} = \sigma/\sqrt{m}$, as $m \to \infty$,*

$$K^v_{\Theta_0^{DGN(m)}} \to K^{(d)}_{FNPF} = d \cdot \sigma^{2(d-1)}_{(m)} H^{(m)}_{FNPF} = d \cdot \sigma^{2(d-1)} H_{FNPF}$$

*Proof.* Let $\Lambda_{\text{FNPF}}^{(m)}$ and $\tau_{\text{FNPF}}^{(m)}$ be quantities associated with DGN$^{(m)}$. We know that $H_{\text{FNFP}}^{(m)} = \Sigma \odot \Lambda_{\text{FNFP}}^{(m)}$. Dropping the subscript FNPF to avoid notational clutter, we have

$$
\begin{aligned}
\left(\sigma/\sqrt{m}\right)^{2(d-1)} \Lambda^{(m)}(s,s') &= \sigma^{2(d-1)} \frac{1}{m^{(d-1)}} \Pi_{l=1}^{d-1} \tau^{(m)}(s,s',l) \\
&= \sigma^{2(d-1)} \frac{1}{m^{(d-1)}} \Pi_{l=1}^{d-1} \left(m\tau(s,s',l)\right) \\
&= \sigma^{2(d-1)} \frac{1}{m^{(d-1)}} m^{(d-1)} \Pi_{l=1}^{d-1} \tau(s,s',l) \\
&= \sigma^{2(d-1)} \Pi_{l=1}^{d-1} \tau(s,s',l) \\
&= \sigma^{2(d-1)} \Lambda(s,s')
\end{aligned}
$$

$\square$

# D    DGN as a Lookup Table: Applying Theorem 5.1 to a pure memorisation task

In this section, we modify the DGN in Figure 2 into a memorisation network to solve a pure memorisation task. The objective of constructing the memorisation network is to understand the roles of depth and width in Theorem 5.1 in a simplified setting. In this setting, we show increasing depth till a point helps in training and increasing depth beyond it hurts training.

**Definition D.1** (Memorisation Network/Task). *Given a set of values $(y_s)_{s=1}^n \in \mathbb{R}$, a memorisation network (with weights $\Theta \in \mathbb{R}^{d_{net}}$) accepts $s \in [n]$ as its input and produces $\hat{y}_\Theta(s) \approx y_s$ as its output. The loss of the memorisation network is defined as $L_\Theta = \frac{1}{2}\sum_{s=1}^n (\hat{y}_\Theta(s) - y_s)^2$.*

| Layer | Memorisation Network |
|---|---|
| Input | $z_\Theta(0) = 1$ |
| Pre-Activation | $q_{s,\Theta}(l) = \sum_j \Theta(i,j,l) \cdot z_{s,\Theta}(j, l-1)$ |
| Hidden | $z_{s,\Theta}(i,l) = q_{s,\Theta}(i,l) \cdot G_s(i,l)$ |
| Final Output | $\hat{y}_\Theta(s) = \sum_j \Theta(1,j,d) \cdot z_{s,\Theta}(j, d-1)$ |

Table 5: Memorisation Network. The input is fixed and is equal to 1. All the internal variables depend on the index $s$ and the parameter $\Theta$. The gating values $G_s(i,l)$ are external and independent variables.

**Fixed Random Gating:** The memorisation network is described in Table 5. In a memorisation network, the gates are *fixed and random*, i.e., for each index $s \in [n]$, the gating values $G_s(i,l), \forall l \in [d-1], i \in [w]$ are sampled from $Ber(\mu), \mu \in (0,1)$ taking values in $\{0,1\}$, and kept fixed throughout training. The input to the memorisation network is fixed as 1, and since the gating is fixed and random there is a separate random sub-network to memorise each target $y_s \in \mathbb{R}$. The memorisation network can be used to memorise the targets $(y_s)_{s=1}^n$ by training it using gradient descent by minimising the squared loss $L_\Theta$. In what follows, we let $K_0$ and $H_0$ to be the NTK and NPK of the memorisation network at initialisation.

**Performance of Memorisation Network:** From Proposition 1.1 we know that as $w \to \infty$, the training error dynamics of the memorisation network follows:

$$\dot{e}_t = -K_0 e_t, \tag{22}$$

i.e., the spectral properties of $K_0$ (or $H_0$) dictates the rate of convergence of the training error to 0. In the case of the memorisation network with fixed and random gates, we can calculate $\mathbb{E}[K_0]$ explicitly.

**Spectrum of $H_0$:** The input Gram matrix $\Sigma$ is a $n \times n$ matrix with all entries equal to 1 and its rank is equal to 1, and hence $H_0 = \Lambda_0$. We can now calculate the properties of $\Lambda_0$. It is easy to check that $\mathbb{E}_\mu[\Lambda_0(s,s)] = (\mu w)^{(d-1)}, \forall s \in [n]$ and $\mathbb{E}_\mu[\Lambda_0(s,s')] = (\mu^2 w)^{(d-1)}, \forall s, s' \in [n]$. For $\sigma = \sqrt{\frac{1}{\mu w}}$, and $\mathbb{E}_\mu[K_0(s,s)/d] = 1$, and $\mathbb{E}_\mu[K_0(s,s')/d] = \mu^{(d-1)}$.

Figure 6: Ideal spectrum of $\mathbb{E}\left[K_0\right]/d$ for a memorisation network for $n = 200$.

Figure 7: Shows the plots for the memorisation network with $\mu = \frac{1}{2}$ and $\sigma = \sqrt{\frac{2}{w}}$. The number of points to be memorised is $n = 200$. The left most plot shows the e.c.d.f for $w = 25$ and the second plot from the left shows the error dynamics during training for $w = 25$. The second plot from the right shows the e.c.d.f for $w = 500$ and the right most plot shows the error dynamics during training for $w = 500$. All plots are averaged over 10 runs.

**Why increasing depth till a point helps ?** We have:

$$\frac{\mathbb{E}\left[K_0\right]}{d} = \begin{bmatrix} 1 & \mu^{d-1} & \dots & \mu^{d-1} & \dots \\ \dots & 1 & \dots & \mu^{d-1} & \dots \\ \dots & \mu^{d-1} & \dots & 1 & \dots \\ \dots & \mu^{d-1} & \dots & \mu^{d-1} & 1 \end{bmatrix} \qquad (23)$$

i.e., all the diagonal entries are 1 and non-diagonal entries are $\mu^{d-1}$. Now, let $\rho_i \geq 0, i \in [n]$ be the eigenvalues of $\frac{\mathbb{E}[K_0]}{d}$, and let $\rho_{\max}$ and $\rho_{\min}$ be the largest and smallest eigenvalues. One can easily show that $\rho_{\max} = 1 + (n-1)\mu^{d-1}$ and corresponds to the eigenvector with all entries as 1, and $\rho_{\min} = (1 - \mu^{d-1})$ repeats $(n-1)$ times, which corresponds to eigenvectors given by $[0, 0, \dots, \underbrace{1, -1}_{i \text{ and } i+1}, 0, 0, \dots, 0]^\top \in \mathbb{R}^n$ for $i = 1, \dots, n-1$. Note that as $d \to \infty$, $\rho_{\max}, \rho_{\min} \to 1$.

**Why increasing depth beyond a point hurts?** As the depth increases the variance of the entries $K_0(s, s')$ deviates from its expected value $\mathbb{E}\left[K_0(s, s')\right]$. Thus the structure of the Gram matrix degrades from (23), leading to smaller eigenvalues.

### D.1 Experiment

We set $n = 200$, and $y_s \sim \text{Uniform}[-1, 1]$. We look at the cumulative eigenvalue (e.c.d.f) obtained by first sorting the eigenvalues in ascending order then looking at their cumulative sum. The ideal behaviour (Figure 6) as predicted from theory is that for indices $k \in [n-1]$, the e.c.d.f should increase at a linear rate, i.e., the cumulative sum of the first $k$ indices is equal to $k(1 - \mu^{d-1})$, and the difference between the last two indices is $1 + (n-1)\mu^{d-1}$. In Figure 7, we plot the actual e.c.d.f for various depths $d = 2, 4, 6, 8, 12, 16, 20$ and $w = 25, 500$ (first and third plots from the left in Figure 7).

**Roles of depth and width:** In order to compare how the rate of convergence varies with the depth, we set the step-size $\alpha = \frac{0.1}{\rho_{\max}}$, $w = 100$. We use the vanilla SGD-optimiser. Note the $\frac{1}{\rho_{\max}}$ in the stepsize, ensures that the uniformity of maximum eigenvalue across all the instances, and the

convergence should be limited by the smaller eigenvalues. We also look at the convergence rate of the ratio $\frac{\|e_t\|_2^2}{\|e_0\|_2^2}$. We notice that for $w = 25$, increasing depth till $d = 8$ improves the convergence, however increasing beyond $d = 8$ worsens the convergence rate. For $w = 500$, increasing the depth till $d = 12$ improves convergence, and $d = 16, 20$ are worse than $d = 12$.