[Reviews · NeurIPS 2020]

Review 1

Summary and Contributions: This work seeks to contribute to our understanding of the importance of gating in deep neural networks (DNNs) by analysing the role of input dependent sub-networks that are a function of ReLU gating masks and the weights of the network. By decoupling the effects of the gating mask, referred to as the *neural path feature* (NPF), from that of the weights of the network, referred to as the *neural path value* (NPV), the analysis is able to more directly establish the importance of gating. The theoretical construction is closely related to the work on the Neural Tangent Kernel (NTK) (Jacot et al., 2018), used to model the error dynamics of DNNs in the infinite width limit. The authors derive a similar kernel referred to as the *neural path kernel* (NPK), a related kernel, but here instead used to model the dynamics of the NPFs (i.e. gating). The main theoretical result of the paper shows that for fixed NPFs and randomly initialised NPVs (weights), the NPK is proportional to the NTK (i.e. the error dynamics of the network directly depend on the gates). Finally, from experimental results with decoupled feature and value finite-width networks compared to infinite-width networks (i.e. GPs), it is argued that the difference in performance between finite and infinite width networks cannot solely be attributed to width, but is in part due to the learning of NPFs, and that most of the important information encoded into a ReLU network during training is in fact stored in the gates.

Strengths: *The work provides an interesting approach to analysing the effect of gating neural networks, with potential for future research directions that might be of interest to the community (i.e. further investigating the neural path kernel). *The analysis provides additional insight into ReLU neural networks as well as the difference between finite and infinite width neural networks. *The decoupled training regime used in the paper could also be of interest to explore further.

Weaknesses: *The angle of analysis is closely related to prior work (Fiat et al., 2019). -- Jonathan Fiat, Eran Malach, and Shai Shalev-Shwartz. Decoupling gating from linearity. CoRR, abs/1906.05032, 2019.

Correctness: In general the work seems correct, both on the side of the theoretical derivations as well as the empirical methodology. ### Minor ### *Figure 2 (\hat{y}_t(x)): This does not seem to be the same as the definition given in the diagram? Should this perhaps get a superscript V to denote it is the output only of the value network? And the output of the DGN is the one given in the diagram?

Clarity: The paper is well written, in fact, I would like to complement the authors on providing a pleasant reading experience (even when having many similarly abbreviated terms that sometimes cause confusion). I specifically found the visual aids (e.g. Figure 1) helpful and appreciated how the authors gradually built their argument. Below are some minor suggested edits or confusions: ### Minor ### *Line 143 (Def 2.2.): I found the use of capitical Lamba and capital A confusing here, and for a moment thought the definition was recursive somehow. *Line 192 (DGN): I would suggest writing DGN out in full once in the main text (it is in the section title, but without the abbreviation next to it.) *Line 237 (.. we expect w/2 gates ..): I found the wording here slightly confusing, perhaps change it to "gates to be on at every layer" or "gates to be active at every layer" *Table 2 (GAP): I would suggest mentioning in the title that GAP stands for global average pooling. This is mentioned in the appendix, but not in the main text. *The citation Arora et al. [2019] appears frequently throughout the text, however there are two possible papers this could be referring to in the reference list. I would suggest adding the usual a,b distinction.

Relation to Prior Work: A related work section is included in the appendix of the paper and seems adequate. However, given the close nature of this work to that of Fiat et al. (2019), I would suggest the authors at least mention this in the main text as well as contrast that work with their own, as is done in the appendix.

Reproducibility: Yes

Additional Feedback: The argument often put forward in prior work for the difference in performance between finite-width networks and infinite-width networks is that the latter essentially ends up being a GP with a fixed feature representation and that the former allows for feature learning, which is often the main motivation behind the use of deep learning. This work shows the importance of gating in ReLU networks, but also that it is specifically the learning of NPFs that explains the source of useful information in the network. However, does this not suggest that the learning of NPFs is simply a manifestation of ordinary feature learning (where the first claim in the paper has pointed out the true source (the gates) but not given any real additional insight on the difference between the two regimes of infinite width vs finite width)? In other words, it was to be expected that learning NPFs would close the gap between the two regimes and the only aspect that is different here is the metric used to measure the phenomenon? I'm not suggesting that this statement is correct and it is possible that I missed a key insight from the work, however, I thought perhaps the authors could comment on this? ### Post-rebuttal update #### I thank the authors for their response, addressing many of the reviewers concerns and for providing more clarity regarding the role of NPF learning as a discriminatory measure for the different learning regimes. I find this work valuable in the sense that it contributes to our understanding of the difference between finite and infinite-width ReLU neural networks, showing that this difference cannot solely be attributed to standard feature learning.


Review 2

Summary and Contributions: Post Rebuttal Update: Thanks to the authors for satisfactorily addressing my concerns in the rebuttal. I have increased my score. Original summary: The authors define two key properties of a ReLU DNN: the Neural Path Feature (NPF) and the Neural Path Value (NPV). The NPF encodes which paths are active and the input features associated with those paths. The NPV encodes the value of all paths from any input to any output (independent of the input). The output of the network can then be expressed as the inner product of the NPF and NPV. The NPF is used to define the Neural Path Kernel (H), while the gradients of NPV are used to define the Value tangent kernel VTK (V). The authors can then formulate the NTK in terms of NPF and VTK, and propose to use the NTK machinery for understanding the importance of gating for generalization. To decouple the effects of learning the gates (that regulate path selection) and the weights that constitute the paths, the authors define a Deep Gated Network (DGN): where a "Feature" network is used only to compute the gates for another "Value" network with identical architecture but separate weights. By extending the NTK analysis to the "Fixed NPF" setting --- where the Feature network weights are kept fixed and Value network weights are trained from a random Bernoulli initialization --- the authors justify their first main claim: that active subnetworks are fundamental entities in ReLU DNNs. Finally, the authors conduct a series of experiments comparing the performance obtained using DGNs and ReLU DNNs. The DGNs are trained under various settings (fixed random Feature weights, fixed but learned Feature weights, etc.). These experiments show that having learned Feature network weights (that determine gating) is important for reaching a performance close to a regular ReLU DNN. The authors use these restuls to justify the second claim of the paper: learning appropriate subnetworks is key for generalization.

Strengths: The main strength of the paper is a analytical treatment of the role of gating in ReLU networks, which could potentially be a useful tool for further analysis. The authors extend NTK theory to quantities that explicitly encode gating information for a network, while previous work had only analyzed the role of gates empirically. The key analytical result is Theorem 5.1, though it relies on Assumption 5.1 (see section on Weaknesses). The second strength of the paper are a set of novel experiments on DGNs, showing explicitly that learning the gating parameters is important to get good generalization (and not for minimizing the training error). The more the gating parameters are allowed to train, the better the generalization performance is.

Weaknesses: Assumption 5.1 does not correspond to how ReLU networks are trained in practice, so I'm not sure how it concretely justifies claim 1. In other words, I don't see how Theorem 5.1 makes the importance of gating more clear when it applies to a hypothetical setting. The claim statement "active subnetworks are fundamental entities" is rather obvious (active subnetworks is what differentiates ReLU networks from linear networks) so I interpret that the goal of the paper is to further identify exactly how they are fundamental. But this exactness applies only under a rather strong assumption.

Correctness: Yes

Clarity: Yes

Relation to Prior Work: Relation to prior experimental work is not made clear. Prior empirical work on importance of active subnetworks investigated their organization in practical settings during learning and their utility as representations of the data, while the experimental study in this work is different in that it examines their importance for generalization specifically, at the same training error, and is also focused on simpler networks (no interaction with regularization etc.).

Reproducibility: Yes

Additional Feedback:


Review 3

Summary and Contributions: The paper studies the role of active subnetworks in deep learning by encoding the gates in neural path features, which are learned during training and this is argued to be a key for datapoint representation and generalization.

Strengths: Very interesting ideas and results. Novel and certainly relevant to NeurIPS.

Weaknesses: The presentation is very weak, so the results are hard to verify to full depth. The experimental evaluation and argumentation is somewhat weak, as only MNIST and CIFAR-10 are considered.

Correctness: The contents seems to be correct.

Clarity: The presentation is very weak.

Relation to Prior Work: Discussion of related work is satisfactory.

Reproducibility: Yes

Additional Feedback: I have read the rebuttal.


Review 4

Summary and Contributions: %%%%%Post rebuttal: I have carefully read the authors' response along with the other reviews. I thank the authors for putting the response together in such a short period of time and I can confirm that I have taken all the information into account when it comes to confirming my final score.%%%%%...The paper is built upon the premises that DNN can be understood through the concepts of neural path feature (NPF) learning and neural path values (NPV). The authors propose that DNNs be considered as having multiple sub-networks and that information is stored in gates. A combination of NPF and NPV characterise what is actually stored in the gates, hence being able to disentangle the pathways from an input to the output towards enhancing interpretability. In addition they provide an empirical explanation for the performance gain of finite width CNN over the pure kernel method based on the exact infinite width CNTK, proposed by Arora et al. 2019. The main contributions of this paper refer to providing an analytical formulation to show how information flows across layes in DNN as sub-networks in various pathways and also how this can improve generalisation.

Strengths: The mathematical formulations are rather strong and detailed, albeit some of them should have been left for the supplementary material. The way authors try to disentangle information propagated through the various layers of DNNs is very interested and they have come up with some very interesting and solid conclusions upon which further work can be done. Generalisation and interpretability are two issues that DNNs show mixed performance depending on the domain in question. Supplementary material provide more context and experiments that clarify issues that remain blurry in the main paper. Nevertheless there is a decent amount of novel components in this paper and is definitely relevant to NeurIPS.

Weaknesses: The analytical part (or most of it) might have been better off as supplementary material and instead include more ablation studies and narrative in the main paper. That would allow more researchers to understand what is being proposed here and potentially consider how this can be adopted or improved upon in the future. In addition I feel that from page 6 onwards comprehending the steps takes a lot of effort as it is hard to follow. Ablation studies should have been explained more as to me the message about how the results have been presented does not come across clearly. In what context is generalisation defined here? Just training on a train set of MNIST and testing on a test set of MNIST? Besides the gain in performance between ReLu and DNPFL and FRNPF appear to be marginal, hence more explanation is needed to quantify what the differences in performances actually mean.

Correctness: The mathematical formulations appear to be correct but it is very hard to check all of them carefully as they are indeed excessive. Some of the narrative included in the supplementary might have been better off in the main paper and vice versa; some of the lemmas and analytical information could have been left for the supplementary material.

Clarity: The paper is arguably quite hard to read and follow. Many formulas and notations throughout, which given the paper length take away from some narrative that is needed to put things into context. It would have been useful if pseudoalgorithms had been included to show in simple terms what is being proposed here. Non-experts in this particular subdomain might find it hard to follow, hence losing some of its potential. Supplementary material help a bit but on the other hand the main paper should suffice to understand the concepts, experiments, ablation studies and conclusions.

Relation to Prior Work: The authors reflect upon previous developments by Arora et al. and build upon limitations identified in the prior work. It is also made clear what the differences, similarities and improvements are as well.

Reproducibility: Yes

Additional Feedback: I think I have covered all aspects I wanted to in the corresponding sections already. May I please ask the authors if they could provide some more information on the generalisation capabilities? Also it might be worth expanding upon the conclusion section to reiterate the main findings and strengthen the position of this paper. Finally I would like to point out that my overall score has considered the issues raised on the clarity side along with the rest of the queries.

[Author Response · NeurIPS 2020]

We thank all the reviewers for their detailed comments. Please find our response to the major comments below (we will
fix all typos/minor comments in the final version).

**Response to Reviewer** 1**:**

`Related work:` We will include comparison with [Fiat et al., 2019] in the main section.

`Feature Learning:` It is true that NPF learning happens when we train standard DNN with ReLU and it is only
natural that it closes the gap. However, NPF learning (interpretation) is different from the standard interpretation of
feature learning, where, the hidden features are learnt in the penultimate layer and the final layer learns a linear classifier
on these features. To see the difference, consider S1={FLNPF, DNPFL, and ReLU-DNN} vs S2 = FRNPF vs S3 =
Infinite width CNTK. S1, S2, S3, all of them generalise (S2 is the least with $67\%$, yet, on a 10 class task like CIFAR-10,
is still way better than random classification accuracy of $10\%$). Both S1 and S2 are finite width, so standard feature
learning happens in both S1 and S2, but, S1 with NPF learning is better ($78\%$ or above in CIFAR-10) than S2 ($67\%$ in
CIFAR-10) with no NPF learning. Thus neither finite width, nor the standard feature learning is useful to explain the
difference between S1 and S2. S2 and S3, both have no NPF learning, yet, S3 generalises better, which can be attributed
to the fact that infinite width ensures better averaging and hence results in a well formed kernel. NPF learning also
happens in the DNPFL setting, which is different from ReLU-DNN. To conclude, *NPF learning is a measure which*
*discriminates/describes the different regimes (i.e., S1, S2, S3) better than the standard feature learning explanation.*

**Response to Reviewer** 2**:**

`Assumption 5.1 and Goal of the paper:` Most analysis of DNNs with ReLU is on what happens at initialisation.
In a DNN with ReLU, NPV and NPF are not statistically independent at initialisation, i.e., Assumption 5.1 does not
hold. However, in the current state-of-the-art analysis, in $w \to \infty$ regime, activations change only at rate of $\sqrt{\frac{1}{w}}$
([Jacot et al., 2019]), i.e., activations/NPFs do not change during training. Hence, though Assumption 5.1 may not hold
exactly, it is *not a strong assumption* to fix the NPFs for the purpose of analysis. Thus, statistically decoupling the NPV
from NPFs is only natural, and furthermore it adds strength: the NTK = const* NPK is *interpretable* in terms of the
active sub-networks (NTK is defined in terms of gradients with no interpretation), which also shows that the active
sub-networks are *fundamental entities arising naturally in the NTK framework.*

Fundamental role of gates is further accomplished by the FLNPF experiments, where we show that by copying gating
information alone, and resetting and training NPV (i.e., $\Theta^{\mathrm{V}}$) from scratch (Assumption 5.1 holds in this case), we can
recover the performance of the DNN with ReLU. Further, in the DNPFL setting, (which is not hypothetical) Assumption
5.1 holds, and the DNPFL does generalise well in the experiments.

`Prior experimental work:` The goal of this paper is not to study the utility of active sub-networks as representations,
but to directly look at the generalisation capability. However, we will move relevant work (example [Srivastava et al.,
2014]) in the appendix to the main body.

`Regularisation:` Similar to [Arora et al., 2019], data-augmentation, batch norm, residual connections, dropout and
other forms of regularisations are avoided. However, studying these in our framework is future work.

**Response to Reviewer** 3**:**

MNIST and CIFAR-10 are used as standard datasets in most analytical works such as ours, see [Arora et al., 2019] for
example.

**Response to Reviewer** 4**:**

`Analytical part in supplementary:` The main contribution of the paper is analytical in nature, and aims at
providing an understanding into the internal workings of a DNN. We believe that it is critical that the setup and
subsequent explanations belong to the main body. While we give details of the experimental setup in the appendix, we
will be happy to move the important points in the main body to improve clarity.

`Explanation of generalisation:` By generalisation we mean performance on test data. Agreed that on MNIST,
all the cases namely FRNPF, DNPFL, FLNPF and ReLU-DNN have marginal performance difference. However, on
CIFAR-10 the difference between FRNPF ($67\%$) and FLNPF, DNPFL, ReLU-DNN (all above $78\%$) is more than $10\%$.
The crucial insight from this work is that mere gating/masking property is enough to give us $67\%$ (on CIFAR-10, this is
non-trivial because a random classifier will only have $10\%$ accuracy), and in addition if the gates also *adapt* during
training (as in standard ReLU-DNN) gives the rest $10\%$. Further, once we have the *learnt* gates, we can reset and learn
NPV from scratch without loss in performance. Thus, the experiments were designed to test the role of gating and we
believe we have extensive experiments to support our claims.

`Expanded conclusion section:` We will make the summary of main contributions more clearer in the conclusion.

[Meta-Review · NeurIPS 2020]

The paper has been well received by all reviews and I find the contribution very interesting as it tries to shed some light to better understand the role of gating in DNNs. The mathematical formulations are strong and detailed as well, which is a plus. There are few typos and some details with the text that should be fixed, the work of Fiat et al. should be clearly cited as well. Some better explanation of the ablation seems that would have been desirable, although I leave to the author to consider whether space allows for better wording.